# Lipidomic profiling of human serum enables detection of pancreatic cancer

Denise Wolrab [1], Robert Jirásko[1], Eva Cífková[1], Marcus Höring[2], Ding Mei[3,4], Michaela Chocholoušková[1], Ondřej Peterka[1], Jakub Idkowiak[1], Tereza Hrnčiarová[1], Ladislav Kuchař [5], Robert Ahrends[6], Radana Brumarová [7], David Friedecký [7], Gabriel Vivo-Truyols[8], Pavel Škrha[9], Jan Škrha[10], Radek Kučera[11], Bohuslav Melichar[12], Gerhard Liebisch [2], Ralph Burkhardt [2], Markus R. Wenk[3,4], Amaury Cazenave-Gassiot [3,4], Petr Karásek[13], Ivo Novotný[13], Kristína Greplová[14,15], Roman Hrstka [15] & Michal Holčapek [1✉]

Pancreatic cancer has the worst prognosis among all cancers. Cancer screening of body fluids may improve the survival time prognosis of patients, who are often diagnosed too late at an incurable stage. Several studies report the dysregulation of lipid metabolism in tumor cells, suggesting that changes in the blood lipidome may accompany tumor growth. Here we show that the comprehensive mass spectrometric determination of a wide range of serum lipids reveals statistically significant differences between pancreatic cancer patients and healthy controls, as visualized by multivariate data analysis. Three phases of biomarker discovery research (discovery, qualification, and verification) are applied for 830 samples in total, which shows the dysregulation of some very long chain sphingomyelins, ceramides, and (lyso) phosphatidylcholines. The sensitivity and specificity to diagnose pancreatic cancer are over 90%, which outperforms CA 19-9, especially at an early stage, and is comparable to established diagnostic imaging methods. Furthermore, selected lipid species indicate a potential as prognostic biomarkers.

[1] Department of Analytical Chemistry, Faculty of Chemical Technology, University of Pardubice, Pardubice, Czech Republic. [2] Institute of Clinical Chemistry and Laboratory Medicine, University Hospital of Regensburg, Regensburg, Germany. [3] Singapore Lipidomics Incubator (SLING), Life Sciences Institute, National University of Singapore, Singapore, Singapore. [4] Department of Biochemistry, Yong Loo Lin School of Medicine, National University of Singapore, Singapore, Singapore. [5] Research Unit for Rare Diseases, Department of Pediatrics and Inherited Metabolic Disorders, First Faculty of Medicine, Charles University and General University Hospital in Prague, Prague, Czech Republic. [6] Department of Analytical Chemistry, University of Vienna, Vienna, Austria. [7] Palacký University Olomouc, Institute of Molecular and Translational Medicine, Olomouc, Czech Republic. [8] Tecnometrix, Ciutadella De Menorca, Spain. [9] Third Faculty of Medicine, Charles University, Prague, Czech Republic. [10] 3rd Department of Internal Medicine, First Faculty of Medicine, Charles University, Prague, Czech Republic. [11] Department of Immunochemistry Diagnostics, University Hospital in Pilsen, Pilsen, Czech Republic. [12] Department of Oncology, Faculty of Medicine and Dentistry, Palacký University and University Hospital, Olomouc, Czech Republic. [13] Clinic of Comprehensive Cancer Care, Masaryk Memorial Cancer Institute, Brno, Czech Republic. [14] Faculty of Medicine, Masaryk University, Brno, Czech Republic. [15] Research Centre for Applied Molecular Oncology, Masaryk Memorial Cancer Institute, Brno, Czech Republic. ✉email: Michal.Holcapek@upce.cz

Non-invasive cancer screening methods based on blood analysis have been intensively investigated in medical research over the last decades[1], with special focus on the detection of early cancer stages. Some cancer types, such as pancreatic cancer[2], do not show specific symptoms, which makes the diagnosis of early stages difficult with established screening methods. Pancreatic ductal adenocarcinoma (PDAC), accounting for 90% of pancreatic cancers, is mostly diagnosed at the late stage resulting in the worst 5-year survival rate (7%) among all cancers[3]. Imaging modalities used to diagnose PDAC in clinical practice include magnetic resonance imaging, computed tomography, endoscopic ultrasound, and positron emission tomography, with accuracy reported in the meta-analysis of 5,399 patients from 52 studies of 90, 89, 89, and 84%, respectively[4]. Invasive procedures, i.e., biopsies, are performed only for the final confirmation of PDAC. Several types of blood tests were considered for PDAC screening[5–7], such as carbohydrate antigen (CA) 19-9 measured alone or with other blood proteins, e.g., carcinoembryonic antigen. The sensitivity and specificity values of CA 19-9 drop for early cancer stages[8], which prevents the applicability for early screening. However, the sensitivity increases for late stage, and therefore CA 19-9 is used for monitoring of cancer treatment. The analysis of circulating tumor DNA, extracellular vesicles, and circulating tumor cells shows a potential for the diagnosis of PDAC and is under investigation. *Kirsten-ras* (*KRAS*) mutation testing is currently used in clinical practice for the epithelial cancer screening (e.g., lung or colorectal cancers)[9] and was evaluated as well for the diagnosis of PDAC using liquid biopsies[10]. However, the sensitivity for *KRAS* mutation testing is low[11], even though this mutation is encountered in more than 90% of PDAC[12]. *KRAS* may be involved in the metabolic reprogramming of fast proliferating tumor cell populations towards elevated glucose and glutamine flows, defined as one of the hallmarks of cancer[13]. Furthermore, the uptake of nutrients in *KRAS* mutated cells can include blood lipids for cell proliferation and survival[14,15]. *KRAS* mutation has been reported to be associated with lipid metabolism in pancreatic cancer cells[16].

Lipids serve numerous functions in human metabolism[17], such as cell membrane constituents, signaling molecules, energy supply, and storage. Changes in lipid concentrations were already reported in other cancer types[18], mostly for cell lines[19], tissues[20], and less frequently for body fluids, too[21].

Here we show differences in serum lipidome concentrations between samples obtained from PDAC patients and healthy controls using mass spectrometry (MS) based approaches followed by statistical analysis.

## Results

**Study design.** Preliminary results showed that the monitoring of single lipid species did not perform well for the differentiation between cases and controls, unlike the multi-analyte approach. Furthermore, lipid species and classes are interrelated, thus it was assumed that the analysis of the lipidome may provide not only molecular biological insights of PDAC but also a more reliable experimental design for clinical diagnostics. The overall methodology is summarized in Fig. 1. Lipid species were quantified by using exogenous lipid class internal standards (IS) added to the serum before the sample preparation (Supplementary Tables 1–4). This allows intra- and inter laboratory comparison because the results are expressed independently from the instrumental signal response. Prepared extracts were analyzed using MS-based approaches, and lipidomic MS data were processed with an in-house script[22] allowing automated lipid identification and quantitation. Finally, the data were statistically evaluated using descriptive

and explorative approaches. All lipid species analyzed with the various MS approaches and within different study phases fulfilled the defined inclusion criteria, i.e., concentrations have to be reported for more than 25% of the samples, otherwise, the lipid species is excluded from the statistical evaluation. This exclusion criterium results in different lipid coverages for individual methods and phases due to natural differences in the sensitivity.

The study was divided into individual phases[23] (Fig. 1) called discovery, qualification, and verification phases, whereby each phase had their own purpose. The sample sets in individual phases were classified into the training and validation sets before applying multivariate data analysis (MDA) to ensure unbiased statistical evaluation. The training set was used to build statistical models, and the validation set for the independent evaluation of the model performance to differentiate samples of cancer patients from healthy controls. The influence of the blood collection tube on the lipidomic analysis was evaluated as a part of the preliminary testing, method optimization, and validation using ultrahigh-performance supercritical fluid chromatography (UHPSFC)/MS[24]. Results showed slightly higher lipid concentrations in serum in comparison to plasma, which yields an enhanced sensitivity. Therefore, serum was used as the sample matrix of choice for the presented PDAC screening study.

**Phase I (discovery).** The discovery phase was a proof-of-concept study with the goal to find differences between serum lipidomic profiles of cases and controls. In total, samples of 262 PDAC patients and 102 healthy controls were analyzed by UHPSFC/MS and shotgun MS, and a limited subset of 64 samples also by matrix-assisted laser desorption/ionization (MALDI)-MS. All methods differ in the detection coverage of lipids, whereby shotgun MS has the highest number of 270 detected lipid species (Supplementary Data 1), followed by UHPSFC/MS with 168 lipid species (Supplementary Data 2), where both methods are based on the positive ion mode. Lipid species belonged to glycerolipids, phospholipids, sphingolipids, and cholesteryl esters for both methods. 42 lipid species from sphingomyelins and sulfatide classes were detected by MALDI-MS in the negative ion mode (Supplementary Data 3). Differences between case and control samples based on the lipidomic profile were visualized by MDA. A partial discrimination between cases and controls was already observed for principal component analysis (PCA) score plots, and the distinct group differentiation was achieved by supervised orthogonal projections to latent structures discriminant analysis (OPLS-DA) models. The influence of gender on the differentiation of case and control samples was investigated (Fig. 2) by performing OPLS-DA for both genders and gender-separated. The accuracy to assign the sample type correctly was slightly better for gender-separated models, therefore gender-separated models were further used in this work, which is in accordance with previously published results[25].

The lipidomic profiling approach for cancer and control samples seems to be independent of the cancer stage because a random distribution of cancer stages is observed in OPLS-DA plots without any clustering (Fig. 3a–d). This finding suggests that the lipidomic profiles differ even for early stage cancer from control samples, which is further verified in the subsequent study phases. The ROC curves and accuracies for training and validation sets were comparable for all methods in the discovery phase (Fig. 3e, f). The most dysregulated lipids are shown in Fig. 3g–j. UHPSFC/MS was used for subsequent studies due to the highest robustness and throughput among the compared methods and also supported by extensive experiences in our group including the full method validation[26] and the stability test for samples collected during one year[24]. The whole sample

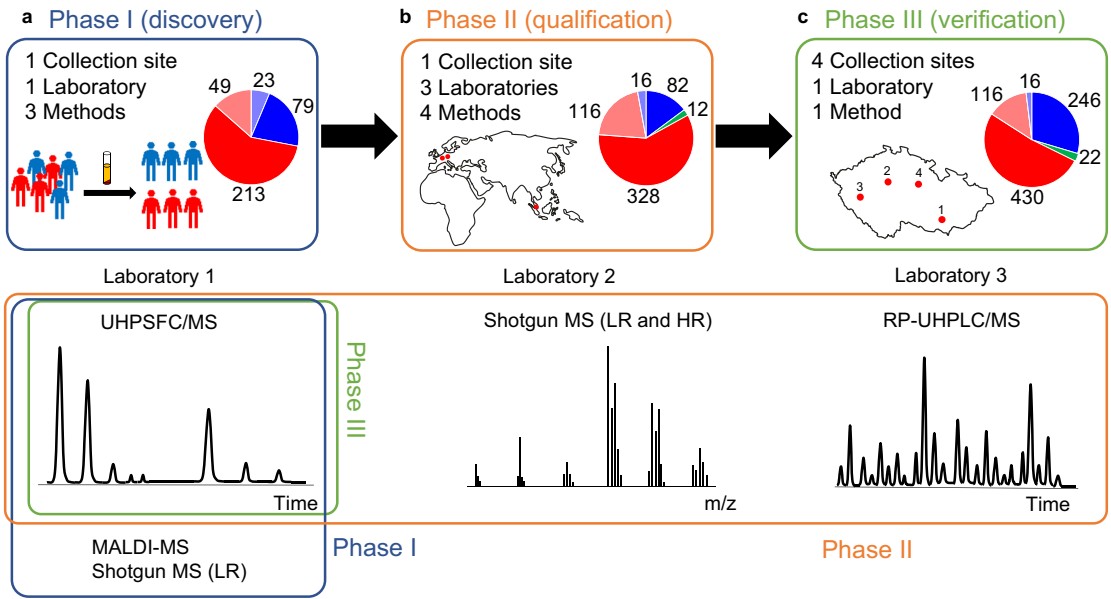

**Fig. 1 Overview of study design for the differentiation of PDAC patients (T, red) from normal healthy controls (N, blue) and pancreatitis patients (Pan, green) based on the lipidomic profiling of human serum using various mass spectrometry-based approaches. a** Phase I (discovery) for 364 samples (262 T + 102 N) divided into training (213 T + 79 N) and validation (49 T + 23 N) sets measured by UHPSFC/MS, shotgun MS (LR), and MALDI-MS. **b** Phase II (qualification) for 554 samples (444 T + 98 N + 12 Pan) divided into training (328 T + 82 N + 12 Pan) and validation (116 T + 16 N) sets measured by UHPSFC/MS, shotgun MS (LR and HR), and RP-UHPLC/MS at 3 different laboratories. **c** Phase III (verification) for 830 samples (546 T + 262 N + 22 Pan) divided into training (430 T + 246 N + 22 Pan) and validation (116 T + 16 N) sets measured by UHPSFC/MS for samples obtained from four collection sites.

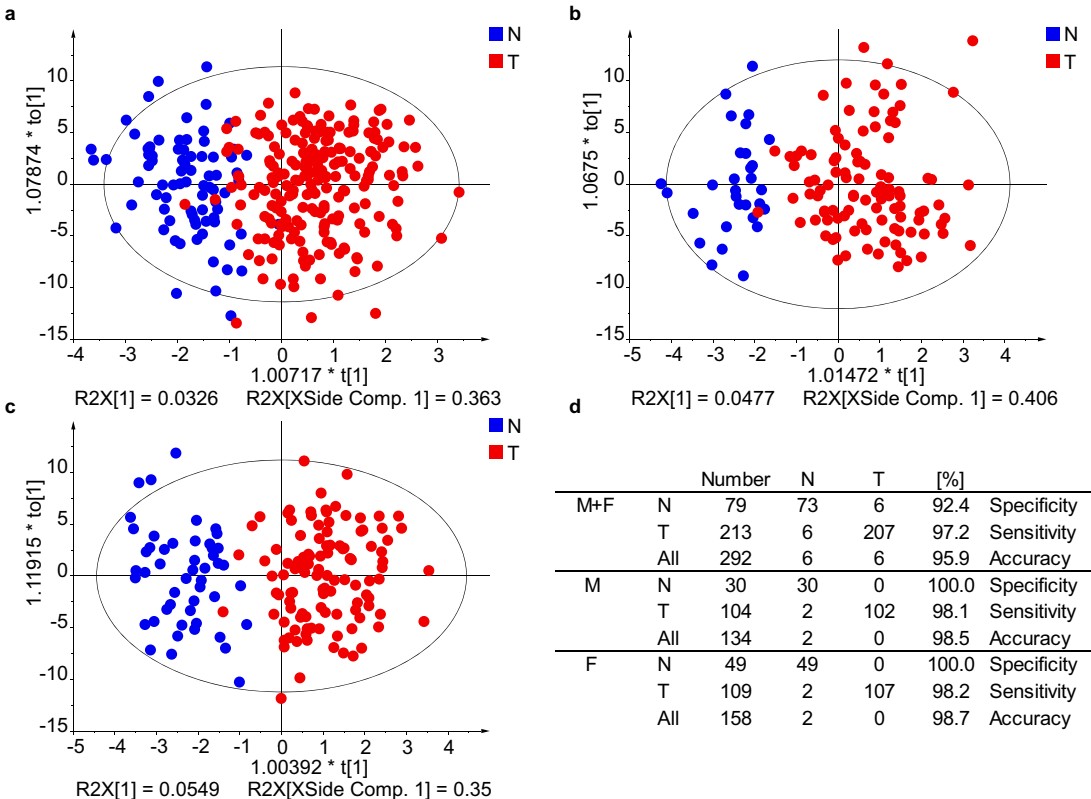

**Fig. 2 Effect of gender separation on the quality of OPLS-DA models used for the differentiation of human serum samples obtained from PDAC patients (T) and healthy controls (N) for the training set using UHPSFC/MS in the Phase I. a** Both genders. **b** Males. **c** Females. **d** Specificity, sensitivity, and accuracy for individual models. Source data are provided as a Source Data file.

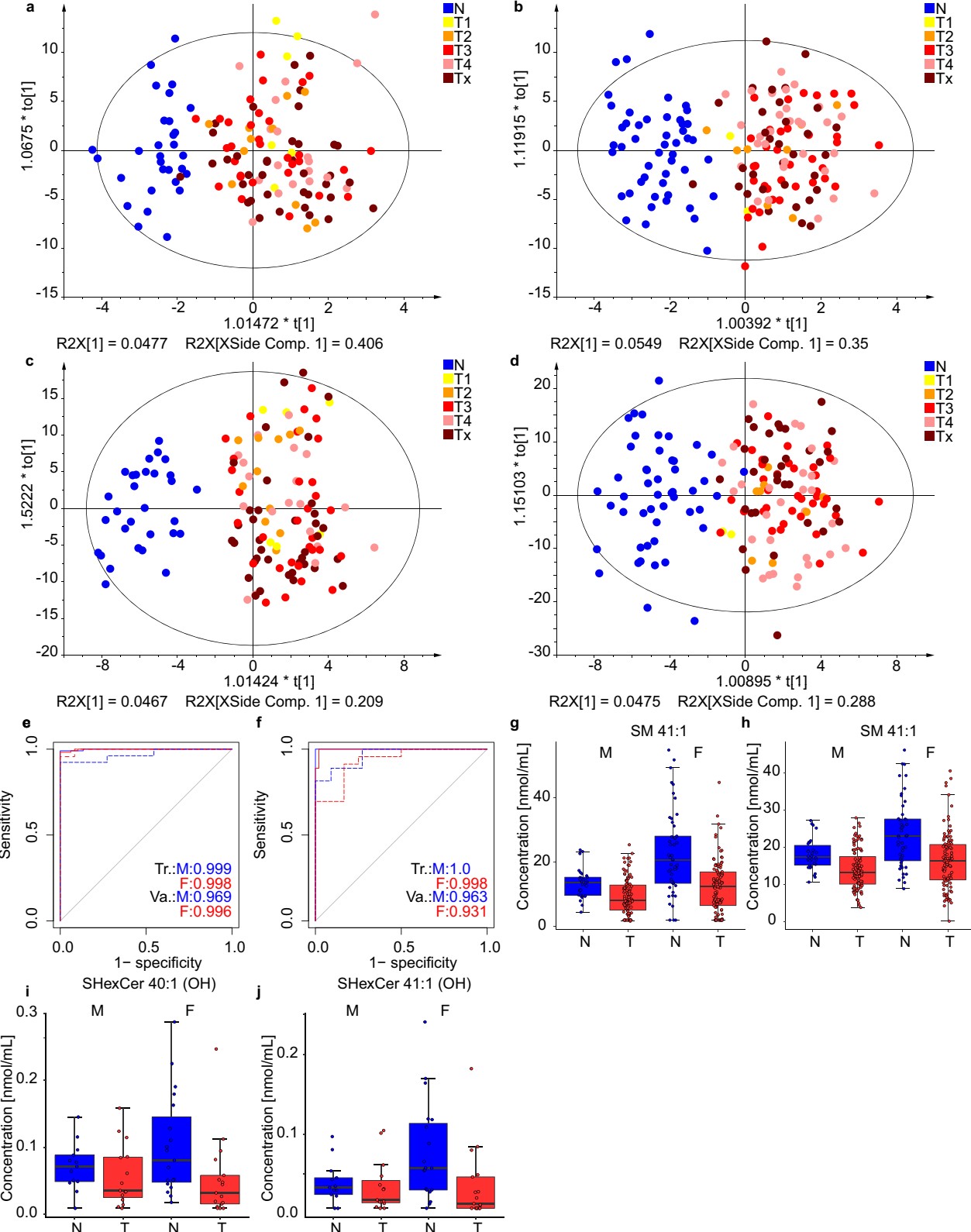

preparation protocol was optimized including the development of quality control (QC) system.

**Phase II (qualification).** The goal of Phase II was to confirm that a similar differentiation of case and control groups can be achieved by other experienced lipidomic laboratories, which should exclude a possible hidden bias for measurements in the single laboratory. Cooperating labs 2 and 3 (details in "Methods" section) had no prescriptions concerning the analytical method used for lipidomic quantitation, so they followed their own established protocols for sample preparation, MS-based measurements, and data processing. The new sample set for Phase II consisted of 554 samples, whereby 344 samples of newly obtained aliquots were from the same volunteers included in Phase I, and

**Fig. 3 Results for the Phase I obtained in lab 1. Individual samples are colored according to tumor (T) stage: T1 - yellow, T2 - orange, T3 - red, T4 - rose, and Tx - brown (information about the stage is not available).** OPLS-DA for males measured **a** with UHPSFC/MS and **c** with shotgun MS for the training set (104 T + 30 N). OPLS-DA for females measured with **b** UHPSFC/MS and **d** shotgun MS for the training set (157 T + 49 N). ROC curves for males (M) and females (F) in training (Tr.) and validation (Va.) sets: **e** UHPSFC/MS, and **f** shotgun MS. Box plots for molar concentration in human serum from PDAC patients (T) and healthy controls (N) for males (M) and females (F): **g** SM 41:1 measured by UHPSFC/MS, **h** SM 41:1 measured by shotgun MS (LR), for both box plots for males (104 T and 30 N) and females (109 T and 49 N), **i** SHexCer 41:1(OH) measured by MALDI-MS, and **j** SHexCer 40:1(OH) measured by MALDI-MS, for both box plots for males (15 T and 14 N) and females (18 T and 19 N). In each box plot, the centerline represents the median, the bounds represent the 1st and 3rd quartile and whiskers span 1.5 fold inter-quartile range from the median. Source data are provided as a Source Data file.

210 samples were from new subjects. The extended cohort was measured by four different MS-based methods (UHPSFC/MS, shotgun MS with low resolution (LR) and high resolution (HR), and reversed-phase ultrahigh-performance liquid chromatography (RP-UHPLC)/MS) (Supplementary Fig. 1). RP-UHPLC/MS allowed the quantitation of 431 lipids (Supplementary Data 4, 5), whereby the lipid species separation is applied due to the hydrophobic interactions of fatty acyls with the nonpolar stationary phase. Shotgun MS is based on the direct sample infusion into MS using specific scan events in case of LR or combined with tandem mass spectrometry (MS/MS) in case of HR. 232 lipids were quantified with shotgun LR-MS (Supplementary Data 6) and 183 lipids with shotgun HR-MS (Supplementary Data 7). For UHPSFC/MS, the lipid class separation was applied, which results in the quantitation of 202 lipid species (Supplementary Data 8). NIST 1950 reference plasma was measured with all methods as well and used for the normalization of lipid concentrations obtained by individual methods separately for males (Supplementary Fig. 2) and females (Supplementary Fig. 3)[27]. The box plots of some of the most dysregulated lipid species (Fig. 4a–c, Supplementary Fig. 2i, j, 3i, and 4a–l) reveal the same pattern and similar normalized concentrations for all methods. The RSD of concentrations of selected lipid species for each sample obtained by four methods (Fig. 4d–f) illustrate the acceptable reproducibility of different quantitation approaches. RSD < 40% for the majority of samples was observed, regardless of the use of different approaches for the sample preparation, IS mixtures, randomization, and lipidomic analysis. The future harmonization of analytical protocols planned within the International Lipidomics Society should further improve the correlation among different laboratories. MDA for individual method data sets from Phase II, such as the ROC curves (Fig. 4g–j), OPLS-DA score plots, and the evaluation of sensitivity, specificity, and accuracy prepared separately for males (Supplementary Fig. 2a–h) and females (Supplementary Fig. 3a–h) were performed. Statistical results show similar outcomes regarding the discrimination of case and control groups for all methods.

**Phase III (verification).** In the timeframe between Phase II and Phase III, UHPSFC/MS method and the sample preparation protocol for the lipidomic analysis were optimized and validated[26,28,29]. Furthermore, the influence of preanalytical and analytical aspects, such as the blood collection tubes[24], lipidomic profile stability in the period of one year for the same volunteers[24], and the influence of the mass spectrometer[30] was systematically investigated. All investigations led to an improved understanding of capabilities of lipidomic profiling for clinical sample screening and illustrated the high reproducibility of UHPSFC/MS for the lipidomic analysis. Phase III aimed at the verification of the applicability of lipidomic profiling for the differentiation of control and cancer samples using the optimized and validated UHPSFC/MS method for the lipidomic analysis of 830 samples (Supplementary Data 9). The sample set consisted of various sample groups obtained from four different blood

collection sites, whereby 554 of 830 samples from clinic 1 correspond to samples from Phase II. The effects of various factors were investigated in addition to PDAC vs. control differentiation, such as pancreatitis, diabetes mellitus, age, cancer stage, and treatment.

The training set included 341 male samples (122 controls and 219 cases, Fig. 5) and 335 female samples (124 controls and 211 cases, Fig. 6). The minor group differentiation was observed in PCA score plots (Figs. 5a and 6a), but OPLS-DA (Figs. 5b and 6b) showed a clear group clustering of PDAC and controls. The influence of the cancer stage was visualized by color codes of samples. No clustering depending on the cancer stage was visible, which indicated that the lipidomic profiling may have a potential for early PDAC detection. The sensitivity, specificity, and accuracy values were overall >94% for the training set and >80% for the validation set (Figs. 5c, 6c, and Supplementary Data 10). The lipid species with the highest concentration differences between case and control samples were visualized by S-plots (Figs. 5d and 6d) and heat maps (Figs. 5e and 6e), whereby lipid concentrations downregulated in case samples are marked in blue, and upregulated lipid species are in red color. Furthermore, statistical tests were performed and lipid species with fold change ≥20%, p-value <0.05 according to the Welch test, and variable importance in the projection (VIP) values >1 were defined as statistical relevant and summarized in Supplementary Data 11–13. Lipid species with p-value < the Bonferroni correction are additionally highlighted and considered as especially statistically significant for the lipidomic differentiation, such as selected sphingolipids, glycerophospholipids, and glycerolipids. However, glycerolipid concentrations may be affected by dietary intake[31], and therefore may be prone to misinterpretation. Consequently, considering statistical parameters (fold change, p-value, and VIP) and excluding exogenous interference, the lipid species SM 41:1, SM 42:1, Cer 41:1, Cer 42:1, SM 39:1, LPC 18:2, and PC O-36:3 were of the highest relevance for the differentiation, which is in accordance with results from Phase I and Phase II.

The effects of the cancer stage and age on the differentiation of case and control samples were further investigated by age-matched controls and early stage PDAC samples classified as T1 and T2 (Fig. 5f). The OPLS-DA model was created for 39 control samples with the average age of 65 ± 4 years and 39 case samples with the average age of 67 ± 4 considering both genders, because age-matched and gender-separated models would result in the insufficient number of samples. The sensitivity, specificity, and accuracy were 97.4% for the differentiation of early cancer stages from control samples, which supports the previous claim on the suitability for early stage PDAC detection and excludes possible bias due to the fact that cancer patients are typically older than healthy controls in many reported studies including this work. Furthermore, the box plots of control, early stage (T1 and T2), late stage (T3 and T4), and pancreatitis (Pan) samples were prepared for statistically most significant lipid species SM 41:1 and Cer 41:1 (Fig. 7b, c). Concentrations measured in cancer samples are downregulated in comparison to control samples

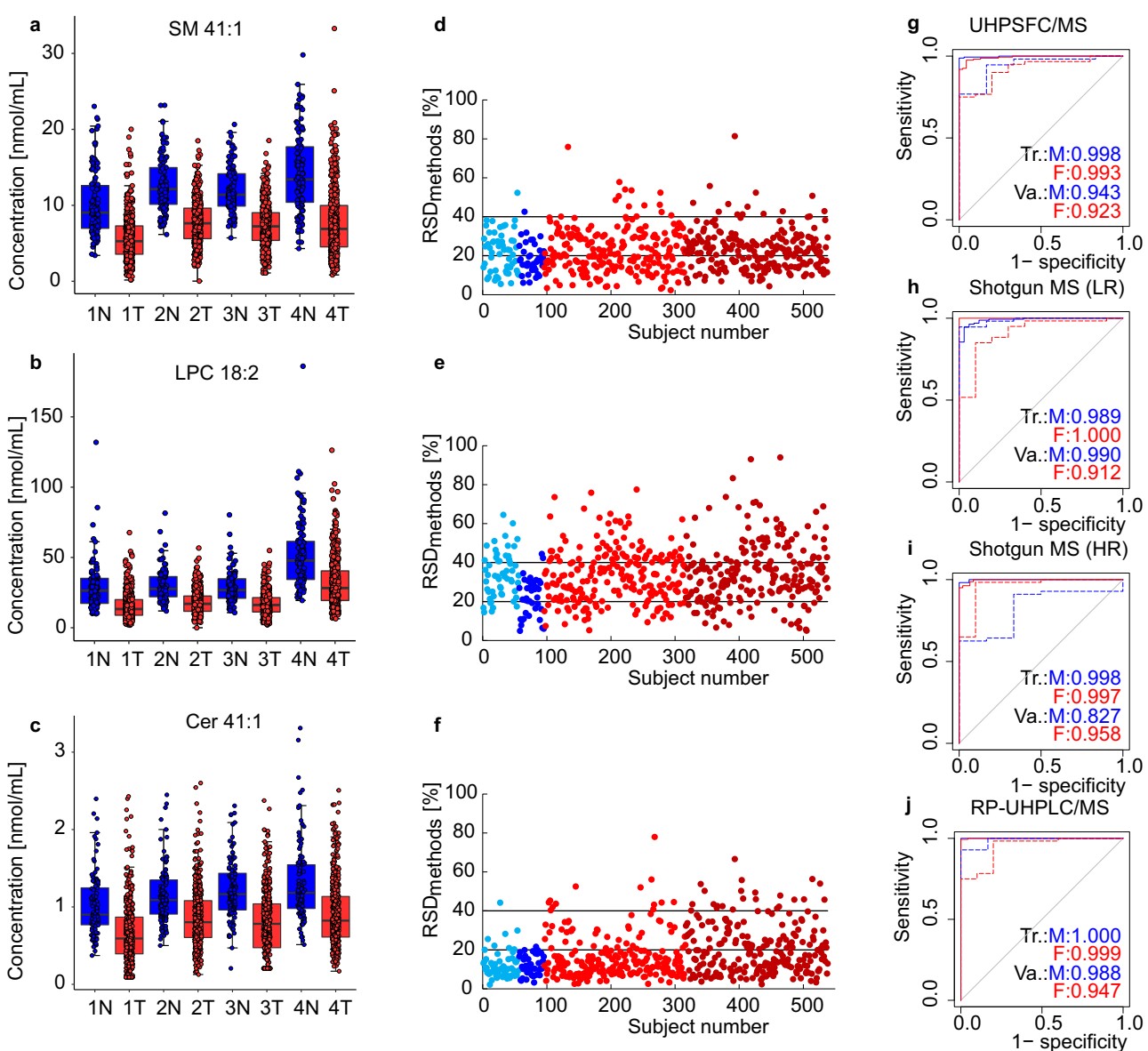

**Fig. 4 Comparison of Phase II results obtained at three different laboratories using four mass spectrometry-based approaches.** Box plots of lipid concentrations normalized to the NIST reference material for samples obtained from PDAC patients (443 T) and healthy controls (95 N) of both genders including both validation and training sets: **a** SM 41:1, **b** LPC 18:2, and **c** Cer 41:1 for UHPSFC/MS (Method 1), shotgun MS (LR) (Method 2), shotgun MS (HR) (Method 3), and RP-UHPLC/MS (Method 4). In each box plot, the centerline represents the median, the bounds represent the 1st and 3rd quartile and whiskers span 1.5 fold inter-quartile range from the median. The RSD of the concentrations obtained by four methods for each sample **d** SM 41:1, **e** LPC 18:2, and **f** Cer 41:1. Color annotation: light blue—control females, blue—control males, red—cancer females, and dark red—cancer males. ROC curves for males (M) and females (F) in training (Tr.) and validation (Va.) sets: **g** UHPSFC/MS, **h** shotgun MS (LR), **i** shotgun MS (HR), and **j** RP-UHPLC/MS. Source data are provided as a Source Data file.

independent of cancer stages, and concentrations in pancreatitis samples are similar to control samples. These results suggest that lipidomic profiling may be applicable for differentiation of pancreatitis from PDAC samples, but the confirmation with a higher sample number of samples within the frame of a prospective study is certainly required. ROC curves for males and females for training and validation sets provided AUC values over 0.90 (Fig. 7a). The effect of age on the most dysregulated lipid species was also visualized for all 830 samples in Phase III (Fig. 7d, e). Lipid concentrations were overall similar for individual age groups, with the exception of slightly elevated lipid concentrations for SM 41:1 and LPC 18:2 for cancer patients younger than 39 years old (Fig. 7d, e), but this observation could be influenced by the smaller number of subjects in this age group.

The diabetes mellitus is connected with a dysfunction of the pancreas, the effect of diabetes on the lipid profiles was investigated by the comparison of lipid concentrations of subjects with and without diabetes mellitus for case and control groups for SM 41:1 (Fig. 7f), where no visible effect was observed.

The overall performance of lipidomic profiling for PDAC screening was compared to the clinical established CA 19-9 biomarker for monitoring the progress of PDAC. Cut-off values for CA 19-9 do not differ for genders, therefore the MDA for lipidomic profiling was performed for both genders as well. Moreover, the comparison was also done for the combination of lipidomic profiling and CA 19-9 to predict sample groups as well as for the recently published CancerSeek method combining the analysis of proteins and ctDNA for cancer screening including

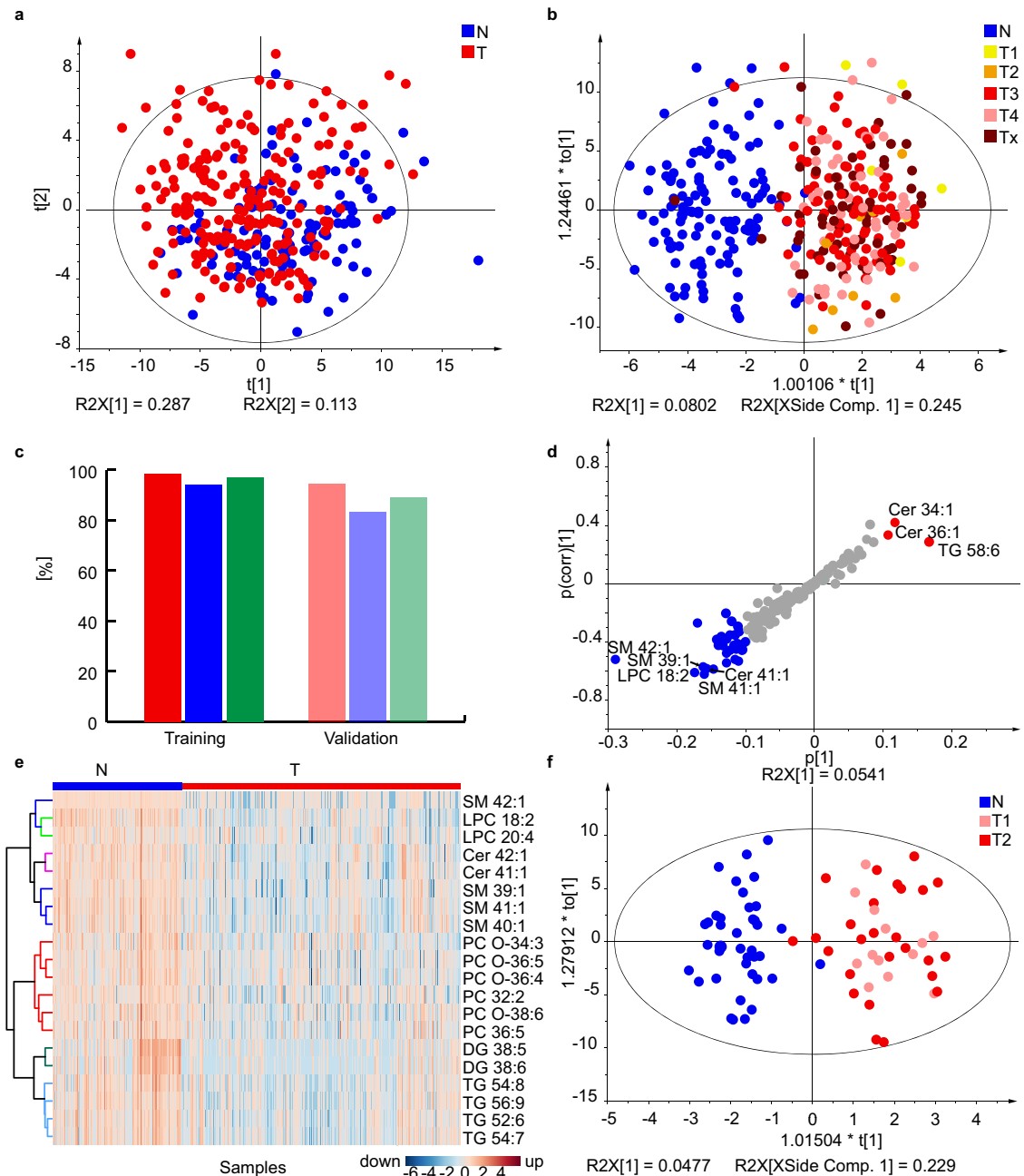

**Fig. 5 Results for the lipidomic profiling of male serum samples from PDAC patients (T) and healthy controls (N) in Phase III. a** PCA for the training set (219 T + 122 N). **b** OPLS-DA for the training set (219 T + 122 N). Individual samples are colored according to tumor (T) stage: T1 - yellow, T2 - orange, T3 - red, T4 - rose, and Tx - brown (information about the stage is not available). **c** Sensitivity (red), specificity (blue), and accuracy (green) for the training (219 T + 122 N) and validation (56 T + 6 N) sets. **d** S-plot for the training set with the annotation of most upregulated (red) and downregulated (blue) lipid species. **e** Heat map for both training and validation sets (275 T + 128 N) using the lipid species concentrations [nmol/mL]. **f** OPLS-DA for early stages T1 + T2, age aligned (mean age is 65 ± 4 years for N and 67 ± 4 for T), and number aligned (39 T + 39 N). This graph includes both genders. Source data are provided as a Source Data file.

PDAC[32]. The ROC curves showed the best performance for the combination of lipidomic profiling and CA 19-9, followed by the lipidomic profiling, the CancerSeek method, and finally the determination of only CA 19-9 (Fig. 7g). The comparison of sensitivity and specificity values for various methods (Fig. 7h) showed that CA 19-9 and CancerSeek yielded higher specificity than sensitivity values[32]. The opposite was observed for lipidomic profiling yielding higher sensitivity than specificity values. The combination of lipidomic profiling and CA 19-9 resulted in increased specificity.

The influence of cancer treatment on the lipidomic profiling was investigated for a small subgroup within the sample set with blood collection before and several days after surgery. MDA plot does not show any return to control group (Fig. 8a), which indicates that PDAC might be a systemic disease with a strong influence on the metabolism, and the tumor removal does result in immediate recovery of lipidomic profile. The box plots for SM 41:1 and LPC 18:2 (Fig. 8b, c) showed that the concentrations mainly decrease after surgery in contrary to control samples (Figs. 5d and 6d). Furthermore, some patients

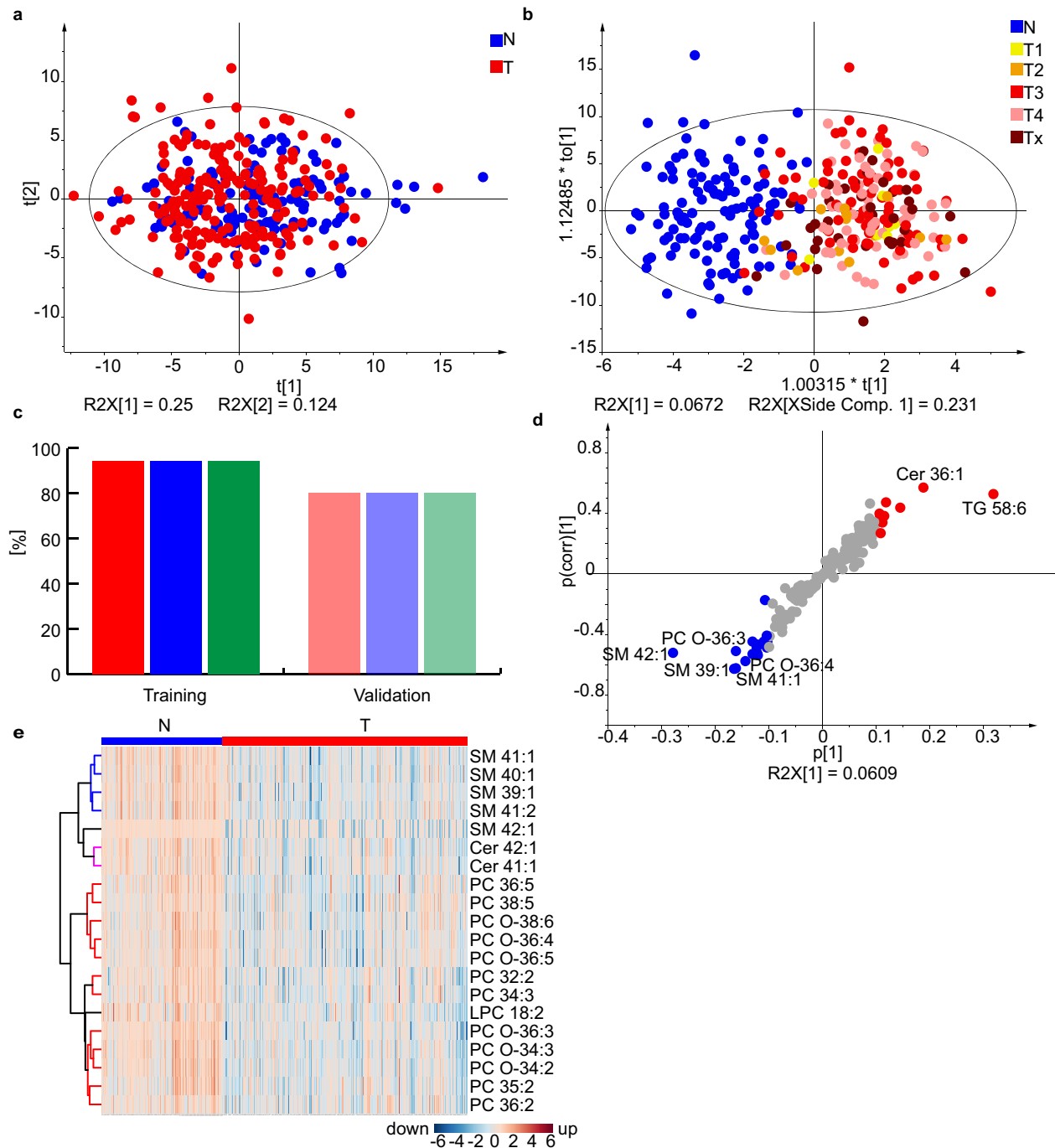

**Fig. 6 Results for the lipidomic profiling of female serum samples from PDAC patients (T) and healthy controls (N) in Phase III. a** PCA for the training set (211 T + 124 N). **b** OPLS-DA for the training set (211 T + 124 N). Individual samples are colored according to their tumor (T) stage: T1 - yellow, T2 - orange, T3 - red, T4 - rose, and Tx - brown (information about the stage is not available). **c** Sensitivity (red), specificity (blue), and accuracy (green) for training and validation sets. **d** S-plot for the training set with the annotation of most upregulated (red) and downregulated (blue) lipid species. **e** Heat map for both training and validation sets (271 T + 134 N) using the lipid species concentrations [nmol/mL]. Source data are provided as a Source Data file.

received medical treatment (e.g., chemotherapy), and samples were collected before and after treatment. No statistically significant effects due to the medical treatment on the lipid profiles were observed (Fig. 8d, e). Furthermore, OPLS-DA models (Fig. 8f, g) were prepared for patients before any treatment, and groups of age-matched healthy controls to exclude any possible biases caused by treatment. The accuracy over 90% and the same patterns of dysregulated lipids show that the actual treatment did not cause relevant changes in lipid profiles.

**Potential for survival prognosis**. The potential of lipids for prognostic purposes was investigated using the data from Phase II for different methods. Lipid concentrations for all samples and the lifetime data were processed with Kaplan–Meier survival analysis for individual methods (Supplementary Data 14 and Supplementary Table 5). Several lipid species showed a statistically significant correlation ($p < 0.05$) with the overall survival (Fig. 9a–c,), such as LPC 18:2, Cer 36:1, PC 32:0, and PC O-38:5. Whereby LPC 18:2 was positively correlated with the survival, in agreement with the previous work[14]. In contrast, Cer 36:1, Cer

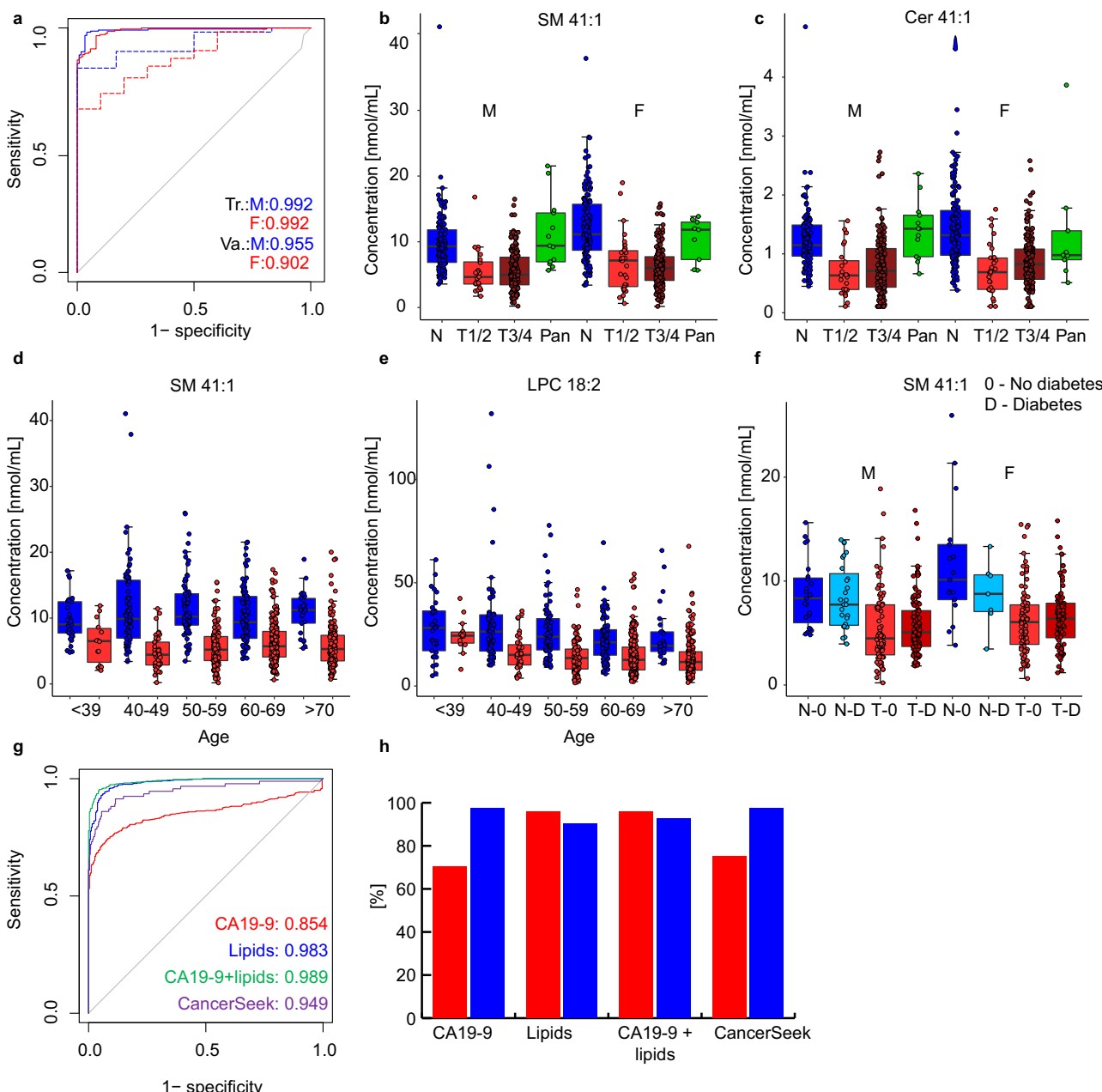

**Fig. 7 Results for the lipidomic profiling in Phase III and investigating the influence of cancer stage, age, and diabetes mellitus. a** ROC curves for males (M) and females (F) in training (Tr.) and validation (Va.) sets. Box plots of lipid molar concentrations normalized to the NIST reference material for: **b** SM 41:1 and **c** Cer 41:1. Only samples with known tumor (T) stage classification were included, where early stage (T1 and T2, 24 males and 30 females) and late stages (T3 and T4, 174 males and 176 females) are summarized and compared to samples of healthy controls (128 males and 134 females) and pancreatitis patients (13 males and 9 females). Comparison of age intervals for control (blue) and cancer (red) samples (**d**) SM 41:1 and **e** LPC 18:2. Box plot investigating the influence of diabetes (**f**) SM 41:1. In each box plot, the centerline represents the median, the bounds represent the 1st and 3rd quartile and whiskers span 1.5 fold inter-quartile range from the median. Comparison of the ROC curves for the samples investigated in Phase III for both genders (**g**) red - CA 19-9, blue - lipidomics, green - combination of CA 19-9 and lipidomics, and purple - CancerSeek[32]. **h** Sensitivity (red) and specificity (blue) for CA 19-9, lipidomics, combination of CA 19-9 and lipidomics, and CancerSeek. Source data are provided as a Source Data file.

38:1, Cer 42:2, PC 32:0, PC O-38:5, and SM 42:2 were negatively correlated with the survival. The gender and treatment did not show any statistically significant effect on the survival probability (Supplementary Fig. 5), but concentrations of CA 19-9 had a strong negative correlation with the survival function (Supplementary Fig. 5). Cox proportional-hazards model was another regression tool used for the visualization of associations among survival time and predictor variables (Fig. 9d), which demonstrated that the concentration of LPC 18:2 higher than median

was positively correlated with survival, while the opposite trend was observed for CA 19-9 and PC O-38:5.

## Discussion
Accurate cancer screening using peripheral blood as a minimally invasive and standardized method is desired in medical healthcare, assuming that early detection of cancer may improve patient outcome. Recently, liquid biopsy based on the analysis of genetic

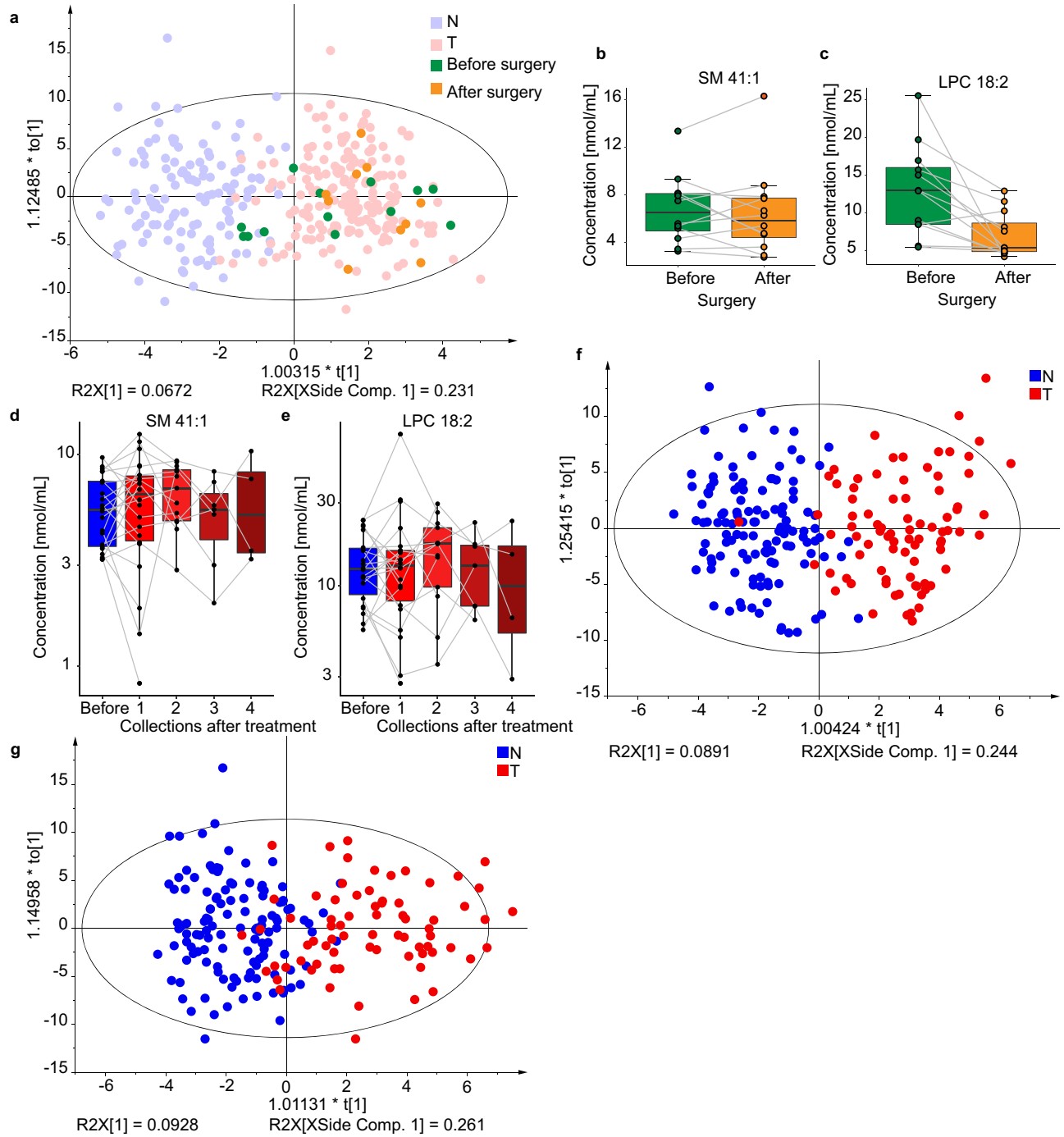

**Fig. 8 Results for the lipidomic profiling of human serum samples for PDAC patients (T) and healthy controls (N) including both genders in Phase III.** Influence of surgery on the lipidomic profile: **a** OPLS-DA for females (211 T + 124 N) using the training set with highlighted samples before (green, $n = 13$) and after (orange, $n = 10$) surgery. Box plots of molar lipid concentrations for paired samples collected before and after surgery for both genders (2 males and 10 females): **b** SM 41:1, and **c** LPC 18:2. Box plots for paired samples collected before ($n = 22$) and after treatment ($n = 22$ for collection 1, $n = 12$ for collection 2, $n = 7$ for collection 3, $n = 4$ for collection 4) for both genders using molar concentrations: **d** SM 41:1, **e** LPC 18:2. In each box plot, the centerline represents the median, the bounds represent the 1st and 3rd quartile and whiskers span 1.5 fold inter-quartile range from the median. OPLS-DA models only for subjects before any treatment or surgery separately for **f** males (83 T + 122 N) and **g** females (72 T + 124 N). Source data are provided as a Source Data file.

mutations[11], ctDNA[10], and proteins[32] in serum or plasma for cancer diagnosis has been intensively investigated, and the results are promising for cancer screening[33]. Furthermore, metabolomics belongs to one of the hot research topics with high expectations in clinical diagnostics. However, the reproducible and comprehensive analysis of small molecules is challenging and often not

achieved in highly complex human body fluid samples[34,35], which is reflected by the overall poor acceptance of metabolomics in comparison to genomics and proteomics despite the enormous research interest. This work is based on a rather complex three-phase concept with the goal to exclude any possible hidden biases and to confirm that the observed changes in serum lipid

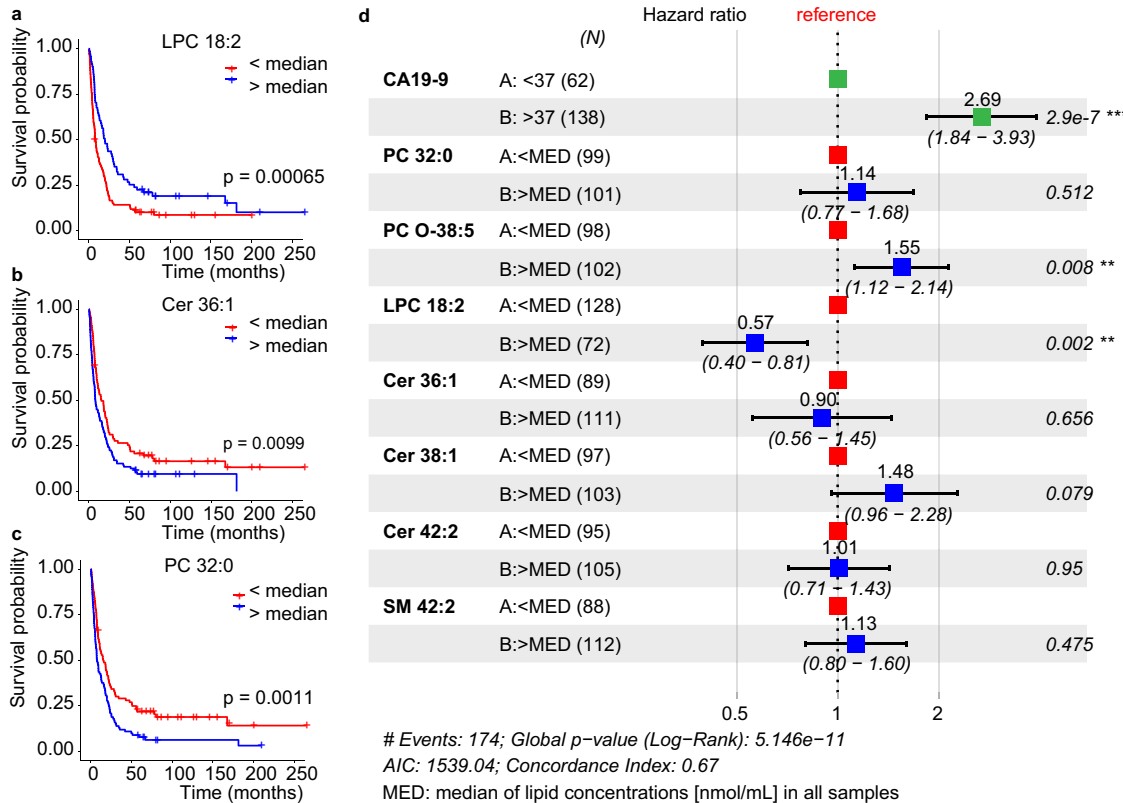

**Fig. 9 Potential of selected lipids for the survival prognosis in Phase II measured by UHPSFC/MS.** Kaplan–Meier Survival plots for: **a** LPC 18:2 ($n = 128$ for binary code 0, and n = 72 for binary code 1), **b** Cer 36:1 ($n = 89$ for 0, and n = 111 for 1), and **c** PC 32:0 ($n = 99$ for 0, and n = 101 for 1) together with the two-sided log-rank test p-value. **d** Cox proportional-hazards model for CA 19-9, PC 32:0, PC O-38:5, LPC 18:2, Cer 36:1, Cer 38:1, Cer 42:2, and SM 42:2. The forest plot illustrates the 95% confidence intervals of the hazard ratios and the corresponding log rank test p values for every parameter are presented. Hazard ratios > 1 indicate poorer survival. Lipid species concentrations normalized to the NIST reference material obtained for all samples in Phase II were converted into the binary code, whereby 0 was set for c < median and 1 for c > median (the median of concentrations was calculated for each lipid species including all samples). Source data are provided as a Source Data file.

concentration are really connected to PDAC, and not influenced by other interfering factors. Our measurements and evaluation consisted of different laboratories (groups in Pardubice, Regensburg, and Singapore), different MS-based workflows (UHPSFC/MS, shotgun LR-MS, shotgun HR-MS, RP-UHPLC/MS, and MALDI-MS), different collection sites (clinics in Brno, Prague, Olomouc, and Pilsen), and sample preparation, IS used for the quantitation, and data processing were done independently by individual laboratories. Regardless of this considerable heterogeneity, we can conclude that reported lipid dysregulation are really statistical relevant for PDAC patients in comparison to healthy controls, and should be reproducible by any laboratory experienced in the quantitative lipidomic analysis. Furthermore, all obtained data sets allow the preparation of MDA statistical models applicable for the differentiation of PDAC patients from controls with relatively high accuracy including early stage PDAC patients. The final confirmation of the applicability of lipidomic profiling to differentiate case from control samples was performed with UHPSFC/MS. The influence of cancer stage, age, diabetes, treatment, and pancreatitis was investigated for 830 samples. 67% of these samples were also included in Phase II. We believe that MS-based lipidomic profiling indicates the potential for early detection of PDAC, but the follow-up confirmatory study and the verification of the clinical utility of such screening are essential before possible implementation into screening programs in individual countries. UHPSFC/MS was selected as the method of choice for further investigations of PDAC screening, but the simple shotgun LR-MS setup may be

also considered for future screening because this configuration is well established in newborn screening.

The comparison of diagnostic performance results revealed that lipidomic profiling can compete with the clinically established method for the monitoring of PDAC progress, CA 19-9, and with CancerSeek (Supplementary Table 6), one of the most promising cancer screening tests published in recent years based on the analysis of ctDNA and proteins. However, CA 19-9 and CancerSeek perform better regarding specificity, important for general population screening, whereby lipidomic profiling performs better regarding sensitivity independent of the cancer stage, which may be of interest for the screening of high-risk individuals. The combination of lipidomic profiling and CA 19-9 improves the diagnostic performance, especially regarding the specificity in comparison to only lipidomic profiling, may be of interest for the establishment of the universal blood screening test.

From the biological point of view, the altered lipid metabolism may originate from tumor cells, tumor stroma, and apoptotic cells. An immune response of the organism may also be involved. All these processes can naturally contribute to the observed cancer lipidomic phenotype. In measurements from all involved laboratories and phases, we observed a clear downregulation of multiple lipid species in the serum of PDAC patients (Fig. 10), such as decreased levels of most very long chain mono-unsaturated sphingomyelins and ceramides. These changes could be linked to the *KRAS*-driven metabolic switch[36]. In this context, alterations in sphingolipids concentrations deserve attention, as

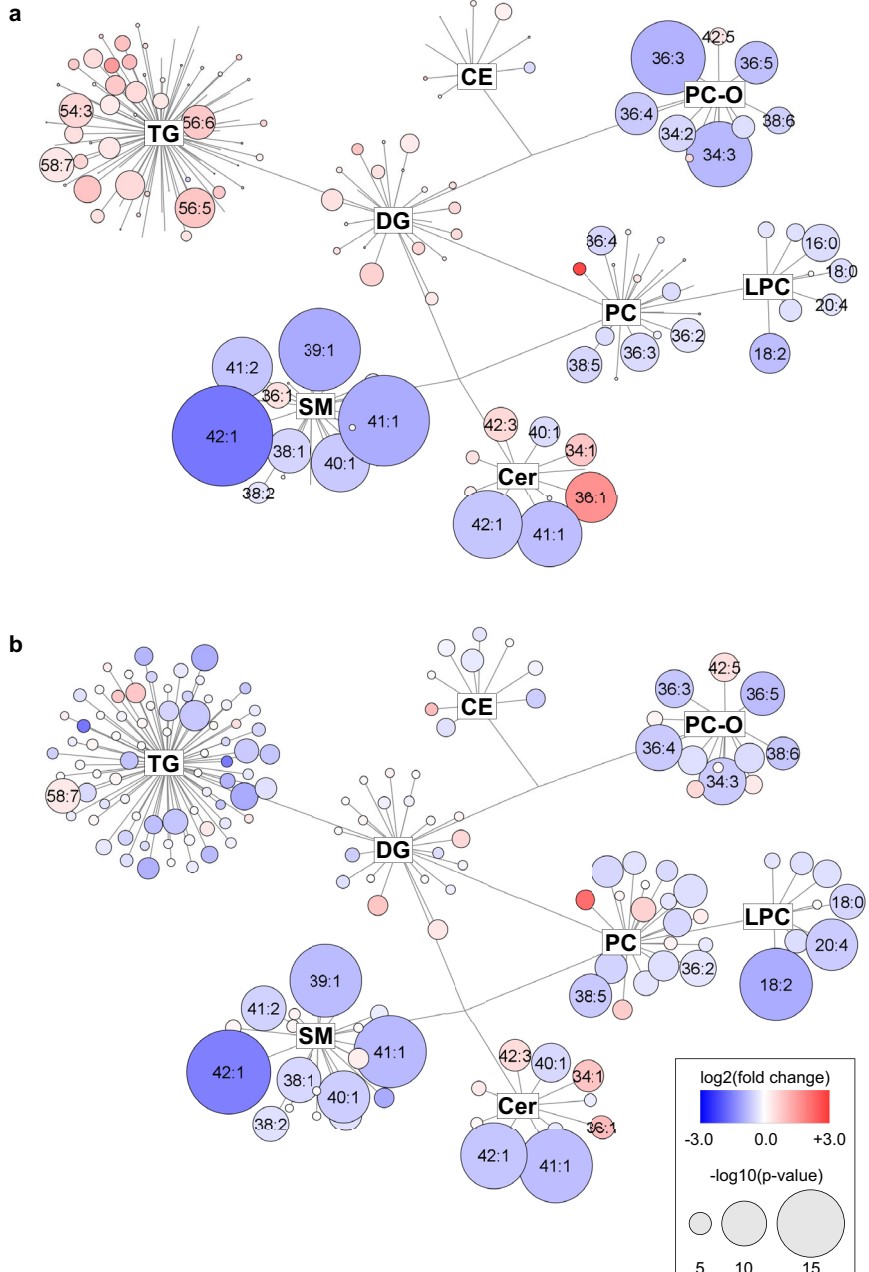

**Fig. 10 Network visualization of the most dysregulated lipid species in PDAC for data from Phase III.** Graphs show lipidomic pathways with clustering into individual lipid classes for **a** males, and **b** females using the Cytoscape software (http://www.cytoscape.org). Circles represent the detected lipid species, where the circle size expresses the significance according to *p*-value, while the color darkness defines the degree of up/downregulation (red/blue) according to the fold change. The most discriminating lipids are annotated. Source data are provided as a Source Data file.

the normal metabolism of sphingomyelins might be necessary to maintain *KRAS* function[37]. Targeted biological investigations are needed to explain the mechanism of lipid alterations in the serum of PDAC patients, but it will require the development of suitable animal models in the future.

In summary, we developed a reproducible, robust, and high-throughput lipidomic profiling approach for the detection of PDAC in human serum, which is applicable for the screening of at least 2000 samples per month on one MS system.

## Methods
**Chemicals and standards**. In lab 1 (University of Pardubice, Czech Republic), solvents for sample preparation and analysis, such as acetonitrile, 2-propanol,

methanol (HPLC/MS grade), hexane, and chloroform stabilized with 0.5–1% ethanol (both HPLC grade), were purchased from either Sigma-Aldrich (St. Louis, MO, USA) or Merck (Darmstadt, Germany), respectively. Mobile phase additives (ammonium acetate, ammonium formate, and acetic acid) were purchased from Sigma-Aldrich. Deionized water was obtained from a Milli-Q Reference Water Purification System (Molsheim, France). Carbon dioxide of 4.5 grade (99.995%) was purchased from Messer Group (Bad Soden, Germany). Non-endogenous lipids used as IS for the quantitative lipidomic analysis were purchased either from Avanti Polar Lipids (Alabaster, AL, USA), Nu-Chek (Elysian, MN, USA), or Merck. Lipid concentrations used for the IS mixture are provided in Supplementary Tables 1 and 2 depending on the employed method, further details for the preparation and dilution of the IS mixture used for UHPSFC/MS measurements were previously published[26]. The NIST SRM 1950 metabolite reference plasma was used as QC sample and for normalization of concentrations between different MS-based methods. Furthermore, a pooled serum sample of PDAC patients and healthy controls were used as QC samples. The lipid annotation used in this

manuscript[38–40] is according to the recommendations of the Lipidomics Standard Initiative (LSI) and given in Supplementary Data 15. The chemicals and standards mentioned above were used for the sample preparation and measurements performed in lab 1.

In lab 2 (University Hospital of Regensburg, Germany), chloroform and 2-propanol were purchased from Roth (Karlsruhe, Germany) and methanol from Merck (Darmstadt, Germany). All solvents were HPLC grade. Ammonium formate and cholesteryl ester (CE) standards were purchased from Sigma-Aldrich (Taufkirchen, Germany). Triacylglycerol (TG) and diacylglycerol (DG) standards were purchased from Larodan (Solna, Sweden) and dissolved in 2,2,4-trimethylpenthane/2-propanol (3:1, $v/v$). Phosphatidylcholine (PC), ceramide (Cer), sphingomyelin (SM), lysophosphatidylcholine (LPC), and lysophosphatidylethanolamine (LPE) standards were purchased from Avanti Polar Lipids (Alabaster, Alabama, USA), and dissolved in chloroform.

In lab 3 (National University of Singapore), chemicals and reagents were obtained from the following sources: ammonium formate, acetic acid, and butanol from Sigma-Aldrich or Merck (Darmstadt, Germany); MS-grade acetonitrile and methanol from Fisher Scientific (Waltham, MA, USA); lipid standards from Avanti Polar Lipids (Alabaster, AL, USA). Ultrapure water (18 MΩ·cm at 25 °C) was obtained from an Elga Labwater system (Lane End, UK).

**Phases of the study**. The study is categorized into three phases in line with the recommendation in the literature[23]: Phase I (discovery), Phase II (qualification), and Phase III (verification). In Phase I, 364 samples were investigated for the lipidomic serum profile differentiation of PDAC patients from healthy controls in the main laboratory (lab 1 - Pardubice) using UHPSFC/MS. For confirmation of results, the samples were again randomly processed and measured with shotgun MS and, for a smaller subset, with MALDI-MS in lab 1. For Phase II, new sample aliquots (554 samples) from the Masaryk Memorial Cancer Institute in Brno were obtained, further re-aliquoted, and distributed among the laboratory at University of Pardubice, Czech Republic (lab 1), the laboratory at University Hospital of Regensburg, Germany (lab 2), and the laboratory at National University of Singapore (lab 3). Each laboratory processed the sample set independently according to their preferred sample preparation method. For the quantitative lipidomic serum profile analysis in all three laboratories, no specifications of the applied MS-based method were provided. The purpose was that the individual laboratories should apply their preferred, optimized, and validated methods for lipidomic analysis. This experimental design is purposely selected to rule out that PDAC differentiation from controls and dysregulation of specific lipids is method-or laboratory-dependent. The following MS-based analytical methods were used for Phase II: UHPSFC/MS (lab 1), shotgun MS with low- and high-resolution (lab 2), and RP-UHPLC/MS (lab 3). The sample preparation protocol and lipidomic analysis were further developed and validated in lab 1 between Phase I and Phase II, and the optimized and validated conditions were applied for Phase II and III[24,26]. Phase III was performed in lab 1 using UHPSFC/MS for the serum lipidomic analysis of samples obtained from different collection sites to verify that lipidomics profiling is diagnostically conclusive and independent of the sample collection site. 554 samples from Phase II are included in 830 samples of Phase III in lab 1.

**Serum samples**. Blood samples were drawn after overnight fasting. For Phase I (364 samples) and Phase II (554 samples), all human serum samples and clinical data were obtained from the Bank of Biological Material (BBM) in Masaryk Memorial Cancer Institute in Brno, approved by the institutional ethical committee, and all blood donors signed informed consent. The sample selection was based on the availability of stored serum samples. The only exclusion criterion for healthy controls (normal, N) was the absence of malignant disease in the life-time history without any other exclusion criteria for other diseases. For all PDAC patients, the disease was confirmed by abdominal computed tomography and/or endoscopic ultrasound followed by needle biopsy or surgical resection. All PDAC patients and healthy controls were of Caucasian ethnicity. The samples were collected from 2013 to 2015. For Phase III (830 samples), serum samples and clinical data were provided by the BBM of Masaryk Memorial Cancer Institute in Brno (554 samples, see Phase II), by the First and Third Faculty of Medicine at the Charles University in Prague (147 samples), by the University Hospital in Pilsen (31 samples) and by the Palacký University and University Hospital in Olomouc (98 samples). All involved institutes provided the ethical approval and signed informed consent for blood collections. Participants did not obtain any compensation for their blood donation. 22 patients with chronic pancreatitis (9 females and 13 males) treated at two outpatient departments were enrolled in this study. The etiology of pancreatitis was either ethanol-induced or recurrent acute pancreatitis. The diagnosis was confirmed by imaging methods (endoscopic ultrasound or endoscopic retrograde cholangio-pancreatography). The overview and detailed description of clinical data and patient characteristics are provided in Supplementary Data 16 and 17. The samples were independently processed for each method used in the study. To avoid biases due to sample collection, sample preparation, and measurements, all samples within the particular phase were processed and measured in the randomized order. The operator had no information about the sample classification during the sample preparation and measurements. The sample sets in all phases were divided into training and validation sets to determine the assay performance using the rigid rule defined before the study that

each 6th sample belongs to the validation set, and the rest constitutes the training set. The sample classification for the training set was disclosed for MDA. The classification of the validation set was disclosed after the final prediction of the validation set.

**Sample preparation**. Briefly, the whole blood was drawn into tubes containing no anticoagulant (Sarstedt S-Monovette, Germany) and incubated at room temperature for 60 min. Then, the samples were centrifuged at 1500 × g for 15 min, the serum was isolated, immediately frozen, and stored at −80 °C until extraction.

The final lipid extraction protocol in lab 1 represents a modified Folch procedure published earlier[24,26]. Human serum (25 μL) and 20 μL of the IS mixture (Supplementary Tables 1 and 2) were homogenized in 3 mL of chloroform/methanol (2:1, $v/v$) for 15 min in an ultrasonic bath (40 °C). When the samples reached ambient temperature, 600 μL of ammonium carbonate buffer (250 mM) was added, and the mixture was ultrasonicated for 15 min. After 3 min of centrifugation (886 × g), the organic layer was removed, and 2 mL of chloroform was added to the aqueous phase. After 15 min of ultrasonication and 3 min of centrifugation, the organic layers were combined and evaporated under a gentle stream of nitrogen. The residue was dissolved in a mixture of 500 μL of chloroform/methanol (1:1, $v/v$) and vortexed. The sample preparation protocol in Phase I was slightly different because only a single extraction was employed without any buffer, with different IS concentrations, and only vortexing instead of ultrasonication.

Finally, the extract was diluted 1:5 with chloroform/methanol (1:1, $v/v$) or 1:20 with the mixture of hexane/2-propanol/chloroform (7:1.5:1.5, $v/v/v$) (Phase I) for the UHPSFC/MS analysis, 1:8 with chloroform/methanol/2-propanol (1:2:4, $v/v/v$) mixture containing 7.5 mM of ammonium acetate and 1% of acetic acid for the shotgun MS analysis, and 1:1 ($v/v$) with methanol for the MALDI-MS.

The lipid extraction in lab 2 was performed according to the Bligh and Dyer protocol[41] in the presence of exogenous lipid species as IS (Supplementary Table 3) using 10 μL of human serum for the extraction. Chloroform phase was recovered by the pipetting robot (Tecan Genesis RSP 150) and vacuum dried. Residues were dissolved in either 7.5 mM ammonium acetate in methanol/chloroform (3:1, $v/v$) (for LR-MS) or chloroform/methanol/2-propanol (1:2:4, $v/v/v$) with 7.5 mM ammonium formate (for HR-MS).

The lipid extraction in lab 3 was performed in a randomized order using the stratified randomization based on the sample group, age, gender, and BMI. The sample extraction was done over three days (~230 samples/day). Human serum samples (~100 μL each) were taken out of −80 °C freezer into a biosafety cabinet and thawed on ice. 10 μL of each serum sample was transferred into 1.5 mL Eppendorf tubes. In addition, 5 μL of each serum sample was pooled together, mixed, and then 10 μL was aliquoted in 59 Eppendorf tubes to constitute batch quality control (BQC) samples. Process blanks (PBLK 1-4) were prepared by aliquoting 10 μL of water into 1.5 mL Eppendorf tubes for extraction control. 10 μL of commercial human plasma was pipetted into 1.5 mL Eppendorf tubes as reference samples (LTR 1-4). 10 μL of NIST SRM 1950 plasma was pipetted into 1.5 mL Eppendorf tubes as additional reference samples (NIST 1-4). The extraction was done on all above-mentioned samples as follows: Add 190 μL of chilled butanol/methanol (1:1, $v/v$) containing IS (Supplementary Table 4) to the samples. Vortex each sample for 10 s and sonicate in ice water for 30 min. Centrifuge at 14,000 relative centrifugal force for 10 min at 4 °C to pellet insoluble. Transfer 140 μL of supernatant into clean vials. Pool 30 μL of lipid extract from each vial (only from samples, not including BQC, NIST, and LTR), mix, and aliquot into 59 vials as technical quality control (TQC) samples. The TQC extract was diluted with chilled butanol/methanol (1:1, $v/v$) to prepare 80, 60, 40, and 20% diluted TQC solutions, which were used to assess the instrument response linearity. The lipid extracts in LC/MS vials were kept in the −80 °C freezer until LC/MS/MS analysis. On the day of analysis, LC/MS vials were taken out of the freezer, thawed at room temperature for 30 min, sonicated in ice-cold water for 15 min, and injected into LC/MS/MS.

**Measurements of CA 19-9**. CA 19-9, a mucin corresponding to the sialylated Lewis (Le)[a] blood group antigen, was quantitatively determined using the electrochemoluminescence immunoassay Elecsys® (Roche, Rotkreuz, Switzerland) according to manufacturer instructions. The CA 19-9 test was performed for all 830 samples from Phase III, whereby 2 outliers were observed and excluded from the study. The repetition of the CA 19-9 measurements for that outlier was not possible due to the limited sample amount. The cut-off value for the CA 19-9 test is 37 U/mL, therefore all values over 37 U/mL were classified as PDAC.

**UHPSFC/ESI-MS conditions (lab 1)**. UHPSFC/MS measurements were carried out on the Acquity Ultra Performance Convergence Chromatography (UPC²) system coupled to the hybrid quadrupole-traveling wave ion mobility time-of-flight mass spectrometer Synapt G2-Si from Waters by using the commercial interface kit (Waters, Milford, MA, USA). The chromatographic settings were used with minor improvements from previously published methods[26,42]. The main difference is that the data were recorded in the continuum mode. The peptide leucine enkephalin was used as the lock mass with the scan time of 0.1 s and the interval of 30 s. The lock mass was scanned but not automatically applied. The noise reduction was

performed on raw files using the Waters compression tool (v4.1), and then data were lock mass corrected as well as converted into centroid data using the exact mass measure tool from Waters. For data preprocessing, the MarkerLynx software (v4.1) from Waters was used. First, the time scan range of each lipid class peak was determined from the base peak intensity (BPI) chromatogram using MassLynx (v4.1), afterwards individual scans were combined from the predefined scan range for each lipid class within a mass range of 50 mDa. Only $m/z$ values with intensities higher than the threshold of 3000 counts were extracted. Obtained tables in MarkerLynx plotting $m/z$ vs. intensity for each lipid class were exported and further processed using LipidQuant software for the identification and quantitation of lipids. LipidQuant (v1.0) is a laboratory-made Excel macro script written in Visual Basic, which helps with the identification of lipid species via comparing measured $m/z$ values with exact $m/z$ values from the embedded database with a mass tolerance of 5 mDa. The identified species were isotopic corrected and quantified by calculating the concentration (nmol/mL) of the lipid species based on the comparison of the intensity of a particular lipid with the intensity of IS of the same lipid class of known concentration. To facilitate the statistical analysis without manipulating the outcome, lipids present in less than 25% of the samples were excluded from the data set, and zero values were replaced by 80% of the minimum for all samples of the corresponding lipid species. The LipidQuant software, instructions on how to use it, details on the scan ranges, exported txt files, as well as raw data, are provided on figshare (https://figshare.com/s/cc087785ca362af7118e).

**Shotgun MS conditions (lab 1).** Shotgun experiments were performed on the quadrupole-linear ion trap mass spectrometer 6500 QTRAP (Sciex, Concord, ON, Canada) equipped with the ESI probe. The AB Sciex Analyst software (v1.6.2) was used for the data acquisition. Characteristic precursor ion (PIS) and neutral loss (NL) scan events were used for the detection of individual lipid classes and previously reported MS settings applied[43]. Then, the data were transferred to the LipidView software (v1.2) for further processing and alignment. For the data analysis, all observed ions in the positive ion mode characterized by $m/z$ values, type of scan, and ion intensities were exported as.txt data file and further processed using the LipidQuant software (v1.0) available on figshare (https://figshare.com/s/b28049603a4f361c818b).

**MALDI-MS conditions (lab 1).** MALDI matrix 9-aminoacridine (Sigma-Aldrich, St. Louis, MO, USA) was dissolved in methanol-water mixture (4:1, $v/v$) to the concentration of 5 mg/mL. Diluted lipid extracts of serum were mixed with matrix (1:1, $v/v$). The volume of 1 μL of extract/matrix mixture was deposited on the target plate using the dried droplet crystallization[44]. A small aliquot of chloroform was applied onto MALDI plate spots before the application of the diluted extract/matrix mixture to avoid the drop spreading. Mass spectra were measured on the high resolution MALDI mass spectrometer LTQ Orbitrap XL (Thermo Fisher Scientific, Waltham, MA, USA) equipped with the nitrogen UV laser (337 nm, 60 Hz) with a beam diameter of about 80 μm × 100 μm. The LTQ Orbitrap instrument was operated in the negative ion mode over a normal mass range $m/z$ 400−2000 with the mass resolution 100,000 (full width at half-maximum definition at $m/z$ 400). The zig-zag sample movement with 250 μm step size was used during the data acquisition. The laser energy corresponds to 15% of the maximum, and 2 microscans/scan with 2 laser shots per microscan at 36 different positions were accumulated for each measurement to achieve a reproducible signal. Each sample (spotted matrix and lipid extract mixture) was spotted five times. The total acquisition time for one sample, including measurements of five consecutive spots, was 10 min. Each measurement was represented by one average MALDI-MS spectrum with thousands of $m/z$ values. The automatic peak assignment was subsequently performed, and $m/z$ peaks were matched with deprotonated molecules from a database created during the identification procedure using the LipidQuant (v1.0) software available on figshare (https://figshare.com/s/cb071be45cd91a7c90e2). This peak assignment resulted in the generation of the list of present $m/z$ of studied lipids with average intensities over all spectra, which was used for further IS or relative normalization and statistical evaluation.

**Shotgun MS conditions (lab 2).** The analysis of lipids was performed by direct flow injection analysis (FIA) using a triple quadrupole (QqQ) mass spectrometer (FIA-MS/MS) and a Fourier Transform (FT) hybrid quadrupole—Orbitrap mass spectrometer (FIA-FTMS). FIA-MS/MS was performed in the positive ion mode using the analytical setup and the strategy described previously[45,46]. The fragment ion of $m/z$ 184 was used for phosphatidylcholines (PC), sphingomyelins (SM)[46], and lysophosphatidylcholines (LPC)[47]. The following neutral losses were applied for: phosphatidylethanolamines (PE) – 141, phosphatidylserines (PS) – 185, phosphatidylglycerols (PG) – 189, and phosphatidylinositols (PI) – 277 (ref. [48]). PE-based plasmalogens (PE-P) were analyzed according to the principles described by Zemski-Berry[49]. Sphingosine-based ceramides (Cer) and hexosylceramides (HexCer) were analyzed using the fragment ion of $m/z$ 264 (ref. [50]).

FIA-FTMS setup was described in detail in previous work[51]. Triacylglycerols (TG), diacylglycerols (DG), and cholesteryl esters (CE) were recorded in the positive ion mode in $m/z$ range 500−1000 for 1 min with a maximum injection time (IT) of 200 ms, an automated gain control (AGC) of 1·10[6], 3 microscans, and a target resolution of 140,000 (at 200 $m/z$). The mass range of the negative ion

mode was split into two parts. LPC and lysophosphatidylethanolamines (LPE) were analyzed in the range $m/z$ 400−650. PC, PE, PS, SM, and ceramides were measured in $m/z$ range 520−960. Multiplexed acquisition (MSX) was used for $[M + NH_4]^+$ of free cholesterol (FC) ($m/z$ 404.39) and cholesterol D7 ($m/z$ 411.43) using 0.5 min of acquisition time with the normalized collision energy of 10%, IT of 100 ms, AGC of 1·10[5], the isolation window of 1 Da, and the target resolution of 140,000. Data processing details were described in Höring et al. using the ALEX software[51,52], which includes the peak assignment procedure and intensity picking. The extracted data were exported to Microsoft Excel 2016 (v16.0.5239.1001) and further processed by the self-programmed Macros available on figshare (https://figshare.com/s/e336bdf3a52f04c2de1f).

Lipid species were annotated according to the shorthand notation of lipid structures derived from MS[2]. For QqQ glycerophospholipid species, the annotation was based on the assumption of even numbered carbon chains only. SM species annotation is based on the assumption that a sphingoid base with two hydroxyl groups is present.

**RP-UHPLC/MS/MS conditions (lab 3).** The RP-UHPLC/MS/MS analysis was performed on the Agilent UHPLC 1290 liquid chromatography system connected to the Agilent QqQ 6495 A mass spectrometer. For the data acquisition, the MassHunter software was used (vB.09.00 – B9037.0) The Agilent Eclipse Plus C18 column (100 mm × 2.1 mm, 1.8 μm) was used for the LC separation. The mobile phases A (30% acetonitrile—20% isopropanol—50% 10 mM ammonium formate in $H_2O$, $v/v/v$ + 0.1% formic acid) and B (90% isopropanol—9% acetonitrile—1% 10 mM ammonium formate in $H_2O$, $v/v/v$ + 0.1% formic acid) were used for both positive and negative ionization. The following gradient was applied: 0 min 15% B, 2.5 min 50% B, 2.6 min 57% B, 9 min 70% B, 9.1 min 93% B, 11 min 96% B, 11.1 min 100% B, 11.9 min 100% B, and 12.0 min 15% B, held for 3 min (total runtime of 15 min). The column temperature was maintained at 45 °C. The flow rate was set to 0.4 mL/min and the sample injection volume was 2 μL.

The spray voltage was set to 3.5 kV in the positive ionization mode and 3 kV in the negative ionization mode. The nozzle voltage was set at 1 kV. The drying gas temperatures were kept at 150 °C. The sheath gas temperature was 250 °C. The drying gas and sheath gas flow rates were 14 and 11 L/min, respectively. The nebulizer gas setting was 20 psi. The iFunnel high- and low-pressure RF were 180 and 160 V, respectively, in the positive ionization mode and 90 and 60 V, respectively, in the negative ionization mode. The MRM list is provided in Supplementary Data 18.

Quantitative data were extracted by using the Agilent MassHunter Quantitative Analysis (QqQ) software (vB.10.00). The data were manually curated to ensure that the software integrated the right peaks. Peak areas of the extracted ion chromatograms peaks for each MRM transition were exported to Microsoft Excel (v1808). Peak areas were normalized to the peak areas of IS using an in-house R (v4.0.0) script and the following packages: here (v0.1), dplyr (v0.8.5), tidyr (v1.0.3), purrr (v0.3.4), readr (v1.3.1), lubridate (v1.7.8), and stringr (v1.4.0). The data quality was assessed using the following criteria, MRM transitions kept for the analysis had to satisfy: coefficient of variation (CoV) measured across the QC injections < 20%, linearity TQC dilution series Pearson $R^2 > 0.80$, signal in processed blanks < 10% of the signal observed in the QC. Data are available at figshare: https://figshare.com/s/1fd10f273b049b93fa24

**Method validation and quality control (lab 1).** The UHPSFC/MS method was validated in line with FDA and EMA guidelines, as previously published[26]. Solvent blanks and QC samples were regularly measured after each 40 samples. For the QC samples, a pooled serum sample and the NIST SRM reference plasma sample were extracted and aliquoted. Furthermore, a mixture of naturally occurring lipid species was used as a system suitability standard. In order to assess the instrumental state, the instrument stability and sample preparation quality, the signal response of selected endogenous lipids, and the IS in all samples were monitored during the whole sequence. The signal responses of selected lipids were plotted against the number of measured samples, which allows the visualization of outliers due to sample preparation or instrumental failures. Typically, a gradual signal drop is observed for the IS caused by contamination of the mass spectrometer over time[24]. Furthermore, PCA for the lipidomic profiles in all samples was performed to review for outliers and clustering of QC samples.

**Statistical analysis.** SIMCA software, version 13.0.3 (Umetrics, Umeå, Sweden) was used to perform the unsupervised PCA with unclassified samples, and the supervised OPLS-DA with the known sample classification. Only scatter plots of the first and second components are presented in PCA score plots. OPLS-DA separates samples based on the known classes and can be used for prediction. Differences in lipid profiles between genders were observed in the Phase I (Fig. 2), therefore data sets for males and females were handled separately. The variables were log-transformed, centered, and scaled (unit variance (UV) or Pareto (Par) scaling) to achieve better performance and model stability. The outliers were evaluated and checked for potential data-entry errors. Logarithmic transformation was applied for each lipid species. Centering relates the relative changes of a lipid species to the average, where UV or Pareto scaling compensates the concentration variance differences for lipid species. The scaling was chosen regarding improved

separation of PDAC patient and control samples and reduced number of outliers without using class information employing PCA. Pareto scaling was superior for UHPSFC/MS, MALDI-MS, low- and high-resolution shotgun MS (lab 2) and RP-UHPLC/MS (lab 3) measurements, and UV scaling for shotgun MS measurements in lab 1 during Phase I. For PCA and OPLS-DA, the number of components was assessed by model fit and prediction ability. In the case of too few components, the differentiation of classes (i.e., health state) is insufficient, while in the case of too many components, the model may be overfitted, resulting in diminished prediction power. The model fit was determined by the evaluation of R2, which describes the variation of variables (lipid species) explained by the model. The insight into the prediction ability of the model is described by Q2 and is estimated using 7-fold cross-validation. PCA plot was evaluated for outliers, errors in measurements, clustering of QC samples as well as for the separation of sample types, i.e., PDAC patients *vs.* healthy controls. Afterwards, OPLS-DA was performed to discriminate between PDAC patients and healthy controls. The number of predictive and orthogonal components for all methods is provided in Supplementary Table 7.

OPLS models were built for the training set for individual methods and validated by the prediction of the validation set using predicted response values. The unpredicted original value of $Y$ is 0, if a human subject is without cancer, and 1 in case of PDAC (binary variable). The predicted response value is continuous and computed using the last model component. Based on the predicted value of $Y$, the sample is classified as non-cancer subject (if predicted $Y \leq 0.5$) or cancer subject (if predicted $Y > 0.5$). A summary of the predicted response values obtained for training and validation sets with the various methods at different clinical phases is provided in Supplementary Data 19 and 20. Depending on the correctly identified healthy and cancer samples, the selectivity, specificity, and accuracy of the model for the training and validation samples were determined (Supplementary Data 10).

To evaluate lipids of statistical significance, a two-sided two sample T-test assuming unequal variances (Welch test) was performed for healthy and cancer samples. $P$-values < 0.05 were considered to indicate the statistical significance. The Bonferroni approach was applied to all p-values for the multiple testing correction. The Microsoft Excel Professional Plus 2016 software (v16.0.5239) was used for these calculations. The summary statistics and average molar lipid concentrations for healthy and cancer samples are summarized in Supplementary Data 11–13 for all methods and study phases. Furthermore, the parameter of variable influence of projection (VIP) was evaluated for each statistical OPLS-DA model using the SIMCA software. Finally, only lipid species with p-values < 0.05, VIP values >1, and fold changes ≥20% for molar concentrations were considered as statistically important and reported in Supplementary Data 11–13. For the visualization of differences in lipid concentrations (up and downregulation) between cancer and control samples, box plots were constructed in R free software environment (v3.6.2) (https://www.r-project.org) using ggplot2 (v3.3.3), ggpubr (v0.4.0), readxl (v1.3.1), dplyr (v1.0.2), and rstatix (v0.6.0) packages. In each box plot, the median was presented by a horizontal line, the box represented 1st and 3rd quartile values, and whiskers stood for 1.5*IQR from the median. Each measurement was plotted as a jittered point value. The receiver operating characteristics (ROC) curves were generated by using the package AUC (v0.3.0) in R.

For verification of the data processing, statistical analysis, and results, data were cross-checked and independently reprocessed or evaluated by applying the online metabolomics platform MetaboAnalyst (v4.0)[53].

The Kaplan–Meier plots and Cox proportional hazards analysis was performed for each of the lipids. The groups of patients with values of particular lipid below the median and above median were compared in terms of survival. The Kaplan–Meier survival analysis plot and the Cox proportional-hazard analysis plots were generated by using the packages survival (v3.2-7), dplyr (v1.0.2), readxl (v1.3.1), and survminer (v0.4.8) in R software. The Cytoscape software (v3.8.0) was used to prepare Fig. 10. The Adobe Illustrator CC 2018 (v22.1, 64 bit) was applied to process graphics and prepare final figures.

**Outlier inspection**. The QC system and the PCA analysis revealed outliers. In Phase I, sample No. 355 was excluded from the UHPSFC/MS data set, and sample No. 210 for the shotgun MS data set, due to the sample preparation failure. The repetition of the sample preparation was not possible due to insufficient serum volume. In Phase II, samples No. 246 and 500 were excluded from the low resolution shotgun MS data set, and samples No. 246 and 409 from the high resolution shotgun MS data set.

**Reporting summary**. Further information on research design is available in the Nature Research Reporting Summary linked to this article.

## Data availability
All data necessary to support the conclusions are available in the manuscript or supplementary information. Source data are provided with this paper. Raw data, instructions for software handling, and the software are deposited at figshare.com:https://figshare.com/s/cc087785ca362af7118e—(UHPSFC/MS; Phase I and Phase II). https://figshare.com/s/e336bdf3a52f04c2de1f—(Shoutgun-MS (LR and HR); Phase II). https://figshare.com/s/cb071be45cd91a7c90e2—(MALDI-MS; Phase I). https://figshare.com/s/1fd10f273b049b93fa24—(RP-UHPLC/MS; Phase II). Source data are provided with this paper.

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

## Acknowledgements

The project was funded by the project 18-12204S (the Czech Science Foundation). L.K. acknowledges the support of institutional program of the Charles University in Prague (UNCE/MED/007). R.H. acknowledges the support of MEYS—BBMRI-CZ LM2018125 (the Ministry of Education, Youth, and Sports of the Czech Republic) and MH CZ—DRO (MMCI, 00209805, the Ministry of Health of the Czech Republic) projects.

## Author contributions

M.H., D.W., R.J., and E.C. prepared the design of experiments. D.W., M.Ch., and O.P. performed sample preparation. D.W. analyzed sample by UHPSFC/MS, E.C. by shotgun MS, and R.J. by MALDI-MS. M.Hör., D.M., G.L., K.G., R.Bu., M.R.W., and A.C.G. performed analysis and data evaluation in cooperating laboratories. E.C. developed the software for data evaluation. D.W., R.J., T.H., E.C., J.I., and G.V.T. processed data and performed statistical analysis. L.K., D.F., R.Br., and R.A. discussed observed changes. R.H., P.K., I.N., P.Š., J.Š., R.K., and B.M. obtained and provided serum samples and corresponding data. M.H., D.W., and R.J. prepared the first draft of manuscript. D.W., R.J., and J.I. prepared figures. M.H. was responsible for funding and supervision of this study. All co-authors read and approved the manuscript.

## Competing interests

M.H., E.C., R.J., and D.W. are listed as inventors on patent EP 3514545 related to this work. All other authors declare no competing interests.
