## [Peer Review File · Nature Communications]

Lipidomic profiling of human serum enables detection of pancreatic cancerREVIEWER COMMENTS

Reviewer #1 (Remarks to the Author); expert on pancreatic cancer and biomarkers:

Wolrab and colleagues present work to design a lipid panel from serum that identifies individuals with pancreatic cancer compared to controls. They use multiple lipid profiling approaches in several laboratories and in a large number of serum samples. The early detection of pancreatic cancer is a worthy area for study and a blood-based approach has advantages over current screening practices, which are limited to only those at high risk for PDAC. Please see comments for consideration below.

The second sentence of the abstract is overstated. We have little data currently to indicate that analysis of body fluids for early detection improves survival times of patients who develop pancreatic cancer.

Sensitivity and specificity data should be provided for CA19-9 in the reported datasets. In the Introduction section, CA19-9 is noted to have sensitivity and specificity of “30-50% for small non-metastatic tumors.” Recent studies suggest sensitivity of 55-65% at specificity of 99% for localized PDAC. Important to know how CA19-9 performs in the reported patients and whether lipids add discrimination to CA19-9. The top-end of a clinical assay normal range is not the cut-off likely to provide high specificity. CA19-9 would need to be measured in controls to best assess the sensitivity and specificity trade-offs in the presented populations.

Recent studies, such as those testing CancerSEEK which combines cfDNA with proteins including CA19-9, are not discussed in the paper. In fact, few other studies are discussed evaluating CA19-9, cfDNA, or protein markers. Additional context is needed for readers to appreciate the field and how the current study contributes new information to this field.

It is unclear what lipids are being included in the models for each of the phases or whether a model is being locked down and then applied to the next phase. Are new models being constructed with each phase, so that the “verification” phase is not really verifying the model from the previous 2 phases? Would also be helpful to know if all measured lipids are being included in the models or just a subset. If a subset, which ones at each phase? More detail on these aspects of the study would be helpful to interpreting the results.

How were the controls enrolled? Are they from clinic patients at primary care practices? They are substantially younger on average than the cases and have substantially less diabetes. It is important to know what base population the controls represent and how that relates to the cases. How are differences in age accounted for in the analyses? Are some of the lipids associated with age?

I don't understand how blood samples would be included in the analyses if they were drawn after surgery or after the patient started chemotherapy to treat their cancer. Surgical resection of the pancreas would presumably have significant impact on lipid levels. Aside from paired pre- and post-surgery measurements to evaluate changes within an individual, why would you want to compare someone with pancreatic cancer s/p Whipple to controls? How is that helpful for disease detection?

At some points in the manuscript, there are descriptions of age-matched analyses that exclude cases post-surgery and try to select cases with early stage disease, but this is difficult to follow, as there are many populations described in the manuscript. More clearly stating up-front what population you are looking to evaluate with a fixed model would be helpful. This way the manuscript reads less like a series of convenience samples and more like you are building to a clinically relevant fixed test in a high yield screening population.

Similar accuracy for late-stage and early-stage disease compared to controls seems unusual. Most commonly sensitivity at a fixed specificity will decline as move from patients with later to earlier stage disease. More attention to this aspect of the study would be warranted.

Can the authors provide a sense of reproducibility of measurements across an analytic run? Were QC samples included for calculation of reproducibility metrics, such as coefficients of variance. When the same lipid was measured by different platforms, were any common samples run on both to allow comparability in measurements or at least ordering of measurements across platforms?

Reviewer #2 (Remarks to the Author); expert on lipidomics:

This is a well-designed study to investigate and compare the lipidome in serum from pancreatic cancer patients and healthy controls. The analysis involved three independent laboratories and applied four different mass spectrometry based analytical platforms. They found distinct group differentiation using supervised orthogonal projections to latent structures discriminant analysis. They suggest that a lipidomic approach is applicable for future screening of pancreatic cancer in high risk groups for early diagnosis.

I have few questions:

1. How were the healthy controls defined? As stated at the beginning of the manuscript "pancreatic cancer does not show specific symptoms making the diagnosis at an early stage difficult and imaging modalities are used to diagnose PDAC". I wonder if the authors can comment on how "healthy" the controls were. Have they been verified by medical imaging? In Fig 3b and 3f, just judged by eye, there are a few healthy controls (blue dots) which appear in the area of the cancer patient group. Do the authors regard these individuals at high risk for developing pancreatic cancers?

2. It is very interesting for analytical readers that the analysis has been done in three different labs with four different methods. The authors stated that "Box plots constructed for the most significantly dysregulated lipid species (Fig. 2e-g, Extended Data Fig. 4i, 5i, and 6) revealed a mutual comparability of molar concentrations from individual laboratories, despite the use of different approaches for sample preparation and lipidomic quantitation". Can the authors comment on how reproducible the measurement of a particular lipid such as SM41:1 was in a given sample across the three labs?

Some minor comments:

3. Were the total of 364 patients & control samples in phase I included in the phase III study?
4. Supplementary table 3, 4 & 5, units for molar concentration are missing.

Reviewer #3 (Remarks to the Author); expert on cancer biomarkers and statistics:

The manuscript summarizes an amazing amount of work performed to discover and validate lipidomic markers of PDAC. In the steps of protein biomarker discovery and validation, they perform the first 3 of discovery, qualification, verification. The study design includes Phase I Discovery (n=364), Phase II qualification (n=554), Phase III verification (n=830). Phases I and II used 3-4 mass spec methods. Specimens were provided from three institutions for the last phase. MS assays were performed at 3 institutions for the second phase. This manuscript describes an immense amount of work. The authors state that the data is available at figshare.com.

Overall, the manuscript is confusing. Factors contributing to the confusion include

- grammar,
 - it isn't clear what decisions are being made at each phase and what conclusions are being carried forward from one phase to the next
 - same for the discovery and validation
 - there are 3 study goals (discovery, qualification, verification), multiple institutions, multiple mass spec methods and it is difficult to track all of these through the paper
- Hopefully the comments that follow will help to shed light on specific areas of confusion.

Strengths:

- The authors performed 3 phases, discovery, qualification, verification
- Figure 1 is really nice. It gives an overall picture of the entire study design.
- The authors appropriately use randomization to avoid biases due to sample collection or preparation order. The training/validation designation was determined prior to any assays being performed.
- Impressively, in all of these hundreds of assays run, less than 10 had to be removed due to assay failure.

Questions and areas for improvement

Human subjects studied/patient population

- The intended screening population is not clear. Eg, do the authors intend this for general population screening or in some high-risk setting?
- The human subject description is buried in the supplemental tables (Supp table 6). In my mind this is one of the most important tables of the paper since it informs the reader about the population studied, and therefore the population to which these inferences can be applied. Thus, it seems it should be moved up rather than being buried in the middle of supplementary information.
- As the specimens were selected based on availability, the sample sets clearly represent a convenience cohort. Convenience cohorts can be severely biased (see Ransohoff 2005 Bias as a Threat... paper). The authors protect against that by including specimens from four sample selection sites. More information would be useful to understand who these patients are, beyond stating that they are otherwise healthy. Eg, are they from patients who were in for a general medical exam, or were they seen in a pancreas clinic, or something else?

Discovery and validation

- It is not clear how many models were created, but it sounds like many.
- Training and validation sets were used in each phase, though it isn't quite clear why and what each validation step was used to validate. Eg, Typically, the discovery phase is used to identify possible candidates, and use of training and validation sets makes sense. Then verification is to confirm that these candidates are indeed differentially abundant. So why was training and validation needed here? Then verification steps include assessment of specificity, etc. I can see how training and validation could be used here, presumably to estimate coefficients in discovery and then fix them to evaluate performance in the validation. It isn't clear though.
- The analyses methods are a curious choice as these are typically used in exploratory analyses. PCA was used for QC and outlier detection. OPLS-DA was used for pdac/control discrimination.
- Both unsupervised and supervised analyses were performed. "The scaling was chosen with regard to improved separation of PDAC and controls and reduced number of outliers..." It is not clear whether the scaling choice was made only in discovery cohorts?
- OPLS-DA (Orthogonal Projections to Latent Structures Discriminant Analysis) was the primary analysis tool. This is largely an exploratory analysis tool.
- 7-fold cross validation was performed. Components (is this principle components? Or variables?) were added while Q2 was increasing. It isn't clear then how the validation was performed.
- The main point here is that it isn't clear what analyses were decided upon before looking at the data, what decisions were made after looking at the data, and how much of that exploratory analysis work "leaked" into the validation cohort analyses.

The model

- A sex effect was identified and so sex-stratified analyses performed.
- Ca19-9 is the best marker available. It is poor (and thus work such as the authors' is needed) but cannot be ignored. How was Ca19-9 considered in the analysis?
- Ideally all of these are put into one model (ca19-9, sex, lipid markers). I don't think this was done (eg sex stratified analyses were done rather than including sex as a covariate). It is important to understand how much discrimination improves above Ca19-9. This is not clear.
- Some unconventional metrics were used. Eg, model fit was assessed by R2 – this does indicate how much variation is explained but doesn't give any indication of metrics of detection. The prediction ability measured by Q2 and was estimated via cross validation – Q2 needs to be defined. It sounds like it may be discrimination or ROC AUC, but that is not clear to those who do not use the software.

Measures like discrimination, sensitivity and specificity, calibration are more meaningful in detection.

- A confidence level of 95% was used for all models. Does this mean variables were included if significant at the 5% level?
- Predicted response was computed using the last model component. What is the last model component?

Other

- There is no discussion section. Some paragraphs in the results seem to belong in a discussion section. A discussion section would be very helpful.
- There are many abbreviations. It would be helpful if there were an index of these, and/or they were defined in figure legends.
- Table 8b. it is not clear what the 1+4+0, etc means. Are the numbers a count of the number of variables or components in the model? This table makes it look like analyses were performed within lab rather than by pooling data across labs for one model?
- The manuscript would benefit from review for grammar and sentence structure.
- Figure 2: shows ROC AUC curves by sex for training and validation, but the line types are the same so you can't tell which is which. Different line types should be used.
- The authors demonstrate that the lipids themselves can be detected by multiple MS platforms, and that they can be assessed quickly and reliably, indicating that they could be used to screen 2000 patients/month on one MS system.
- Analyses were cross checked in MetaboAnalyst. So all of the QC, OPLS-DA analyses, cross validation, etc were re-done in two different software packages?

Response to reviews

First of all, we would like to kindly thank all reviewers for careful reading and insightful comments how to improve the manuscript. We have done our best to respond adequately to all queries step-by-step.

Reviewer #1 (Remarks to the Author); expert on pancreatic cancer and biomarkers:

Wolrab and colleagues present work to design a lipid panel from serum that identifies individuals with pancreatic cancer compared to controls. They use multiple lipid profiling approaches in several laboratories and in a large number of serum samples. The early detection of pancreatic cancer is a worthy area for study and a blood-based approach has advantages over current screening practices, which are limited to only those at high risk for PDAC. Please see comments for consideration below.

The second sentence of the abstract is overstated. We have little data currently to indicate that analysis of body fluids for early detection improves survival times of patients who develop pancreatic cancer.

- The sentence was re-written to avoid the overstatement.

Sensitivity and specificity data should be provided for CA19-9 in the reported datasets. In the Introduction section, CA19-9 is noted to have sensitivity and specificity of “30-50% for small non-metastatic tumors.” Recent studies suggest sensitivity of 55-65% at specificity of 99% for localized PDAC. Important to know how CA19-9 performs in the reported patients and whether lipids add discrimination to CA19-9. The top-end of a clinical assay normal range is not the cut-off likely to provide high specificity. CA19-9 would need to be measured in controls to best assess the sensitivity and specificity trade-offs in the presented populations.

- CA19-9 was measured in all case and control samples from Phase III using an electro-chemo luminescence immunoassay Elecsys® (Roche, Rotkreuz, Switzerland). The cut-off value for CA19-9 was defined as 37 U/mL, therefore all values >37 U/mL were classified as PDAC samples. The performance of CA19-9 was compared to lipidomic profiling for the reported cohort. Furthermore, the combination of CA19-9 and lipidomic profiling was investigated, as discussed in the Results chapter for Phase III.

Recent studies, such as those testing CancerSEEK which combines cfDNA with proteins including CA19-9, are not discussed in the paper. In fact, few other studies are discussed evaluating CA19-9, cfDNA, or protein markers. Additional context is needed for readers to appreciate the field and how the current study contributes new information to this field.

- A discussion and comparison of CancerSeek and CA19-9 was added to the manuscript. In general, the whole structure of the first manuscript draft was changed from Letter to Article format, which allows an extended discussion and comparison to previous findings in order to get a better estimation of lipidomic profiling performance to other methods.

It is unclear what lipids are being included in the models for each of the phases or whether a model is being locked down and then applied to the next phase. Are new models being constructed with each phase, so that the “verification” phase is not really verifying the model from the previous 2 phases? Would also be helpful to know if all measured lipids are being

included in the models or just a subset. If a subset, which ones at each phase? More detail on these aspects of the study would be helpful to interpreting the results.

- Different lipidomic methods slightly vary in their lipid coverage due to the different measurement principles. Furthermore, the small differences in the method sensitivity may occur depending on the state of mass spectrometer, which could result in the fact that some low abundant lipid species may not be detected in all cases. Consequently, the lipidomic profiles could slightly differ, especially for low abundant lipids, depending on the employed method and instrument. However, the same inclusion criteria were applied (25% of the samples needed to have a concentration value) for all methods. The same approach was used throughout the whole work. A separate model was created using the predefined training set for each method in each phase, and the diagnostic performance was evaluated by predicting the validation set. Results indicate that lipidomic profiling combined with MDA is applicable for differentiation of case and control samples regardless of the MS based method. The manuscript was re-written and more details for each phase together with the purpose are provided. We believe that results and discussion part should be clarified now.

How were the controls enrolled? Are they from clinic patients at primary care practices? They are substantially younger on average than the cases and have substantially less diabetes. It is important to know what base population the controls represent and how that relates to the cases. How are differences in age accounted for in the analyses? Are some of the lipids associated with age?

- Generally, all samples were chosen based on the availability at the clinical site with predefined inclusion/exclusion criteria for this study. For all PDAC patients, the disease was confirmed by abdominal computed tomography and/or endoscopic ultrasound followed by needle biopsy or surgical resection. All PDAC patients and healthy controls were of Caucasian ethnicity. The only exclusion criterium for healthy controls was the absence of malignant diseases in the life-time history, but no other exclusion criteria for other diseases were applied. Therefore, the overall sample set is a convenience cohort, assuming that differences between cases and controls are universal and applicable for the screening of high-risk individuals. The overview and detailed description of clinical data and patient characteristics are provided in Supplementary Tables 1 and 2. The control samples are overall younger than the case samples, the influence of age on the lipid concentrations/profiles were investigated by using age-matched samples for MDA (Fig. 5) and preparing box plots for case and control samples divided into age intervals (Fig. 7). Furthermore, the influence of diabetes mellitus on the lipid concentrations for the most dysregulated lipid species were investigated (Fig.7). Findings indicate that neither age (except for very young case samples) nor diabetes is significantly influencing the lipidomic profiles. Results and discussion on the influence of age and diabetes were added to the results section.

I don't understand how blood samples would be included in the analyses if they were drawn after surgery or after the patient started chemotherapy to treat their cancer. Surgical resection of the pancreas would presumably have significant impact on lipid levels. Aside from paired pre- and post-surgery measurements to evaluate changes within an individual, why would you

want to compare someone with pancreatic cancer s/p Whipple to controls? How is that helpful for disease detection?

- The authors fully agree that case samples after surgery or treatment are not suited for diagnostic purposes, and this was not the purpose of this sub-aim. Our sub-aim was to investigate the potential of lipidomic analysis for prognostic purposes and the influence of any treatment on lipid concentrations. However, the treatment and surgical resection did not show significant effects on lipid concentrations (Fig. 8).

At some points in the manuscript, there are descriptions of age-matched analyses that exclude cases post-surgery and try to select cases with early stage disease, but this is difficult to follow, as there are many populations described in the manuscript. More clearly stating up-front what population you are looking to evaluate with a fixed model would be helpful. This way the manuscript reads less like a series of convenience samples and more like you are building to a clinically relevant fixed test in a high yield screening population.

- The structure of the manuscript was changed and significant editing was performed, hopefully leading to a better understanding.

Similar accuracy for late-stage and early-stage disease compared to controls seems unusual. Most commonly sensitivity at a fixed specificity will decline as move from patients with later to earlier stage disease. More attention to this aspect of the study would be warranted.

- A more detailed discussion on the influence of the cancer stage on the differentiation of case and control samples using lipidomic profiling was added (Fig. 5 and Fig. 7).

Can the authors provide a sense of reproducibility of measurements across an analytic run? Were QC samples included for calculation of reproducibility metrics, such as coefficients of variance. When the same lipid was measured by different platforms, were any common samples run on both to allow comparability in measurements or at least ordering of measurements across platforms?

- In phase II, 550 case and control samples plus NIST SRM 1950 plasma as QC were measured by 4 different methods in 3 different laboratories using different sample preparation protocols. The RSD of each sample from the 550 measured with the 4 methods was calculated and plotted for selected species in Fig. 4. For most of samples, RSD < 40% was observed for lipid species concentrations measured with all methods. Considering the different nature of analysis method, different extraction protocols, the complexity of the sample and different laboratories, it was concluded that lipidomic profiling is reproducible.

Reviewer #2 (Remarks to the Author); expert on lipidomics:

This is a well-designed study to investigate and compare the lipidome in serum from pancreatic cancer patients and healthy controls. The analysis involved three independent laboratories and applied four different mass spectrometry based analytical platforms. They found distinct group differentiation using supervised orthogonal projections to latent structures discriminant analysis. They suggest that a lipidomic approach is applicable for future screening of pancreatic cancer in high risk groups for early diagnosis.

I have few questions:

1. How were the healthy controls defined? As stated at the beginning of the manuscript "pancreatic cancer does not show specific symptoms making the diagnosis at an early stage difficult and imaging modalities are used to diagnose PDAC". I wonder if the authors can comment on how "healthy" the controls were. Have they been verified by medical imaging? In Fig 3b and 3f, just judged by eye, there are a few healthy controls (blue dots) which appear in the area of the cancer patient group. Do the authors regard these individuals at high risk for developing pancreatic cancers?

- All samples were chosen based on the availability at the clinical site. The absence of malignant diseases in the life-time history of healthy control patients was defined as the sample inclusion criteria, but no other exclusion criteria for other diseases were applied. However, no imaging methods were applied to confirm that control samples were without PDAC. Therefore, the overall sample set is a convenience cohort, assuming that differences between case and control is universal and applicable for the screening of high-risk individuals. The overview and detailed description of clinical data and patient characteristics are provided in Supplementary Tables 1 and 2. Whether the control samples classified as cancerous samples, belong to a group at high-risk to develop pancreatic cancer cannot be stated due to lack of follow up clinical data. This needs to be confirmed in the future within a prospective study investigating thousands of samples belonging to different risk groups.

2. It is very interesting for analytical readers that the analysis has been done in three different labs with four different methods. The authors stated that "Box plots constructed for the most significantly dysregulated lipid species (Fig. 2e-g, Extended Data Fig. 4i, 5i, and 6) revealed a mutual comparability of molar concentrations from individual laboratories, despite the use of different approaches for sample preparation and lipidomic quantitation". Can the authors comment on how reproducible the measurement of a particular lipid such as SM41:1 was in a given sample across the three labs?

- In phase II, 550 case and control samples plus NIST SRM 1950 plasma as QC were measured by 4 different methods in 3 different laboratories using different sample preparation protocols. The RSD of each sample from the 550 measured with the 4 methods was calculated and plotted for selected species in Fig. 4. For most of the samples, RSD < 40% were observed for the lipid species concentrations measured with all methods. Considering the different nature of analysis method, different extraction protocols, the complexity of the sample and different laboratories, it was concluded that lipidomic profiling is reproducible. The RSD of SM 41:1 concentrations in all samples for all platforms (4*550) is 51.3% and reflects the biological, pathological, sample preparation and

method variance. However, this RSD was not further investigated, as the sample reproducibility of a lipid concentration within different laboratories is of higher importance and more meaningful.

Some minor comments:

3. Were the total of 364 patients & control samples in phase I included in the phase III study?
 - The majority of the 364 patients from Phase I were included in Phase III, depending on the sample amount available at the clinical site. For Phase III, new samples from various clinical sites as well as new aliquots were obtained, for patients already investigated in Phase I. The text was modified for better clarity.
4. Supplementary table 3, 4 & 5, units for molar concentration are missing.
 - The unit is nmol/mL serum and was added to the Supplementary tables.

Reviewer #3 (Remarks to the Author); expert on cancer biomarkers and statistics:

The manuscript summarizes an amazing amount of work performed to discover and validate lipidomic markers of PDAC. In the steps of protein biomarker discovery and validation, they perform the first 3 of discovery, qualification, verification. The study design includes Phase I Discovery (n=364), Phase II qualification (n=554), Phase III verification (n=830). Phases I and II used 3-4 mass spec methods. Specimens were provided from three institutions for the last phase. MS assays were performed at 3 institutions for the second phase. This manuscript describes an immense amount of work. The authors state that the data is available at figshare.com.

Overall, the manuscript is confusing. Factors contributing to the confusion include

- grammar,
- it isn't clear what decisions are being made at each phase and what conclusions are being carried forward from one phase to the next
- same for the discovery and validation
- there are 3 study goals (discovery, qualification, verification), multiple institutions, multiple mass spec methods and it is difficult to track all of these through the paper
- **The format style of the manuscript was changed from Letter to Article format. This allows to provide more detailed explanations of individual phases, hopefully contributing to better clarity. The individual phases are now discussed in individual chapters. The manuscript was edited and reviewed by all co-authors, which should lead to improved grammar and spelling.**

Hopefully the comments that follow will help to shed light on specific areas of confusion.

Strengths:

- The authors performed 3 phases, discovery, qualification, verification
- Figure 1 is really nice. It gives an overall picture of the entire study design.
- The authors appropriately use randomization to avoid biases due to sample collection or preparation order. The training/validation designation was determined prior to any assays being performed.
- Impressively, in all of these hundreds of assays run, less than 10 had to be removed due to assay failure.

Questions and areas for improvement

Human subjects studied/patient population

- The intended screening population is not clear. Eg, do the authors intend this for general population screening or in some high-risk setting?
- **Samples were chosen based on the availability at the clinical site. For all PDAC patients, the disease was confirmed by abdominal computed tomography and/or endoscopic ultrasound followed by needle biopsy or surgical resection. All PDAC patients and healthy**

controls were of Caucasian ethnicity. The absence of malignant diseases in the life-time history of healthy control patients was defined as sample inclusion criteria, but no other exclusion criteria for other diseases were applied. No imaging methods were applied for control samples to confirm that control patients are medical normal. Therefore, the overall sample set is a convenience cohort, assuming that differences between case and control is universal and applicable for the screening of high-risk individuals. The combination of the analysis of CA19-9 and lipidomic profiling leads to improved selectivity and sensitivity, which may allow the future testing of clinical utility for the screening of high-risk individuals.

- The human subject description is buried in the supplemental tables (Supp table 6). In my mind this is one of the most important tables of the paper since it informs the reader about the population studied, and therefore the population to which these inferences can be applied. Thus, it seems it should be moved up rather than being buried in the middle of supplementary information.

- The tables were shifted to the front of the Supplementary Information (Supplementary Table 1 and 2).

- As the specimens were selected based on availability, the sample sets clearly represent a convenience cohort. Convenience cohorts can be severely biased (see Ransohoff 2005 Bias as a Threat... paper). The authors protect against that by including specimens from four sample selection sites. More information would be useful to understand who these patients are, beyond stating that they are otherwise healthy. Eg, are they from patients who were in for a general medical exam, or were they seen in a pancreas clinic, or something else?

- A more detailed description about the sample selection was provided in the manuscript.

Discovery and validation

- It is not clear how many models were created, but it sounds like many.

- In fact, for each method in each phase a model for males and females was created for the training set (Phase I: 3x2 methods x gender, Phase II: 4x2 methods x gender, and Phase III: 1x2; resulting in 16 models). The validation set for each method and model was used for evaluating the prediction performance to correctly identify a sample group on an independent data set. Furthermore, patient characteristics such as gender, age, cancer stage, diabetes and the effect of normalization were investigated by MDA, resulting in even more models. The purpose of the separate model-building in each phase of the study was to check the most important variables (lipids) will be consistent across different data sets. Also, the first phases of the study (Phase I and Phase II) were rather small, so the final model, identifying the most important lipids, needed to be built on Phase III data to have the most precision of the estimates in the components. In order to reduce the complexity, the whole manuscript was restructured and the purpose as well as the aim of individual phases is explained in more detail in individual chapters.

- Training and validation sets were used in each phase, though it isn't quite clear why and what each validation step was used to validate. Eg, Typically, the discovery phase is used to identify possible candidates, and use of training and validation sets makes sense. Then verification is to confirm that these candidates are indeed differentially abundant. So why was training and

validation needed here? Then verification steps include assessment of specificity, etc. I can see how training and validation could be used here, presumably to estimate coefficients in discovery and then fix them to evaluate performance in the validation. It isn't clear though.

- As stated in the response above, the purpose of early phases (Phase I and Phase II) was not to develop the model, which will be validated on the Phase III data.

The applicability of lipidomic profiling for the differentiation of case and control samples was observed in the discovery phase using 3 methods. All three methods differ in their characteristics and lipid coverage. UHPSFC/MS was chosen as method of choice for further investigations, due to the high robustness and throughput of this method.

In order to improve the differentiation of case and control samples by lipidomic profiling as well as to evaluate the advantages and limitations of UHPSFC/MS for the analysis of biological samples, several parameters were investigated before continuation with the inter-laboratory study (Phase II). The optimization of the sample preparation protocol, the full method validation according to the guidelines of EMA and FDA (Wolrab et al., *Anal. Bioanal. Chem.* 412 (2020) 2375), the influence of the used blood collection tube on the lipidomic profile and the one year stability of the lipidomic profile for multiple collections were investigated (Wolrab et al., *ACA.* 1137 (2020) 74-84). All these investigations led to an improved lipid coverage and method sensitivity of the lipidomic UHPSFC/MS analysis.

Phase II aimed the evaluation of the impact of using different sample preparation protocols, analysis methods or operating laboratories on the lipidomic profile. Comparable lipid concentrations and similar differentiation performance of case and control samples were obtained, even though the lipidomic profiles differ.

Phase III was used to confirm the applicability of the optimized method for lipidomic profiling to differentiate case and control samples for a bigger sample set. Samples from various collection sets were analyzed and the influence of biological parameters on the prediction performance was investigated, such as cancer stage, age, diabetes, and pancreatitis. The evaluation of the models by an independent data set (validation set) was therefore necessary and should lead to a higher confidence of the whole study.

The applicability of lipidomic profiles and not individual lipid species (biomarkers) to differentiate case and control samples is in accordance with the inter-correlation of lipid metabolism. Even though, some lipid species, like the sphingolipids SM 41:1, SM 42:1 and Cer 41:1, are statistically more relevant for the differentiation of case and control samples, confirmed by all 3 Phases.

The manuscript was restructured and re-written for better clarity.

- The analyses methods are a curious choice as these are typically used in exploratory analyses. PCA was used for QC and outlier detection. OPLS-DA was used for pdac/control discrimination.
- Lipidomic profiling is a multidimensional approach based on the analysis of many lipid species, which are inter-correlated to each other, which makes the explorative analysis using MDA applicable. It is not intended to reduce the lipid amount and perform biomarker

analysis, where other statistical models such as logistic regression or classification trees may be better suited.

- Both unsupervised and supervised analyses were performed. “The scaling was chosen with regard to improved separation of PDAC and controls and reduced number of outliers...” It is not clear whether the scaling choice was made only in discovery cohorts?

- The pareto scaling was applied for all methods and phases to build the OPLS-DA models (except for shotgun/MS in the discovery phase). Pareto scaling was chosen based on the discovery cohort (Phase I) and then only confirmed as suitable also for data from other study phases. The text was modified for better clarity.

- OPLS-DA (Orthogonal Projections to Latent Structures Discriminant Analysis) was the primary analysis tool. This is largely an exploratory analysis tool.

- MDA was used due to the multidimensional character of lipidomics profiling. As the amount of parameters in lipidomic profiling is huge and the choice between those cannot be made by the biological rationale. Multidimensional method was used instead of other classification methods (logistic regression, classification trees), which cannot handle large amount of parameters on limited sample size. The OPLS-DA models were created with the training set and the validation set was used for prediction as an independent data set. This is common practice for supervised analysis approaches, to avoid misjudgment of the predictive performance to correctly classify sample groups, i.e., due to overfitting.

- 7-fold cross validation was performed. Components (is this principle components? Or variables?) were added while Q2 was increasing. It isn't clear then how the validation was performed.

- The 7-fold cross validation is an embedded tool from SIMCA software. The data set is splitted into 7 sample groups, whereby each 7th observation is excluded for each cross validation step. This approach is according to Eastment et al [Eastment, H. and Krzanowski, W., Crossvalidatory choice of the number of components from a principal component analysis, *Technometrics* 24 (1982) 73-77.]. First the data for the observations are excluded to get the loading vectors and then the data for variables to get the score vectors. The excluded data set is then predicted and compared with actual values. The procedure is repeated until all samples were excluded once. The cross validation is performed to evaluate if a component is considered significant and accordingly the number of components for a model is selected, when the autofit option is chosen (which was the case for all models).

This cross validation is an automated tool to define the components of a model and necessary to build the models. However, the validation of the model performance was evaluated by the prediction on testing dataset using the already created model (from the training set).

- The main point here is that it isn't clear what analyses were decided upon before looking at the data, what decisions were made after looking at the data, and how much of that exploratory analysis work “leaked” into the validation cohort analyses.

- The training and validation set were defined prior statistical analysis. The creation of the MDA model follows a standard operating procedure, such as the application of log-transformation, pareto scaling, centering, and applying the autofit function to define the number of components in SIMCA (for PCA and OPLS-DA). The models were always prepared on training set, locked and then validated on the validation set. The choice of the models was not made based on the performance on the validation set.

The model

- A sex effect was identified and so sex-stratified analyses performed.
- Ca19-9 is the best marker available. It is poor (and thus work such as the authors' is needed) but cannot be ignored. How was Ca19-9 considered in the analysis?
- During the revision, CA19-9 was measured for almost all remaining case and control samples included in Phase III. Only a few samples are not available anymore. The diagnostic performance of CA19-9 and lipidomic profiling for the same data set was compared to each other. Furthermore, CA19-9 and lipidomic profiling was combined and the sensitivity and specificity values were compared to individual approaches. Data are provided in the Results chapter.
- Ideally all of these are put into one model (ca19-9, sex, lipid markers). I don't think this was done (eg sex stratified analyses were done rather than including sex as a covariate). It is important to understand how much discrimination improves above Ca19-9. This is not clear.
- All parameters (CA19-9, sex and lipid profiles) were put into one model and the performance was compared to individual approaches as well as to the recently published method called CancerSeek, as a promising diagnostic tool based on the analysis of ctDNA and proteins. Generally, the sensitivity was higher for lipidomic profiling and the specificity was higher for CA19-9 and CancerSeek. Data are shown in Fig.7 and discussed in the Results chapter.
- Some unconventional metrics were used. Eg, model fit was assessed by R2 – this does indicate how much variation is explained but doesn't give any indication of metrics of detection. The prediction ability measured by Q2 and was estimated via cross validation – Q2 needs to be defined. It sounds like it may be discrimination or ROC AUC, but that is not clear to those who do not use the software. Measures like discrimination, sensitivity and specificity, calibration are more meaningful in detection.
- R2 and Q2 are parameters describing the cross validation and model parameters for creating the PCA or OPLS-DA models in SIMCA. The ROC curves, sensitivity and specificity values were determined by the predictive response values computed from the created MDA models and are used as main parameters for model performance evaluation.
- A confidence level of 95% was used for all models. Does this mean variables were included if significant at the 5% level?
- A confidence interval of 95% describes the uncertainty of parameters and was designated before data processing. It gives the probability that 95% means of samples will be within

the confidence interval (eclipse in the score plot). The level of significance of 5% was not used to reduce the number of parameters in the model. The parameters in the model are automatically chosen by the method by combining the most important ones into components of the model.

- Predicted response was computed using the last model component. What is the last model component?
- The predicted response is a value ranging from 0 to 1, whereby values between 0.5 and 1 are classified/predicted as a predefined sample group i.e. cancer and values from 0 to 0.5 as control group. The value is always estimated by the last component of the OPLS-DA model, which is automatically calculated. The exact formula for the last component is however not available in the software.

Other

- There is no discussion section. Some paragraphs in the results seem to belong in a discussion section. A discussion section would be very helpful.
- A discussion session was added.
- There are many abbreviations. It would be helpful if there were an index of these, and/or they were defined in figure legends.
- The manuscript was carefully checked that all abbreviations were defined.
- Table 8b. it is not clear what the 1+4+0, etc means. Are the numbers a count of the number of variables or components in the model? This table makes it look like analyses were performed within lab rather than by pooling data across labs for one model?
- The table describes the number of components used for individual models, important for the reproduction of the models. Allowing readers to repeat and reevaluate the models. Not a single model for all methods and phases was used. A detailed description of the aim of individual phases and the corresponding models are provided in the Results session.
- The manuscript would benefit from review for grammar and sentence structure.
- The manuscript was re-written and all co-authors performed grammar and spelling check.
- Figure 2: shows ROC AUC curves by sex for training and validation, but the line types are the same so you can't tell which is which. Different line types should be used.
- The line types for the training and validation set were changed for better differentiation.
- The authors demonstrate that the lipids themselves can be detected by multiple MS platforms, and that they can be assessed quickly and reliably, indicating that they could be used to screen 2000 patients/month on one MS system.
- The RSD of lipid concentrations within each sample measured with 4 methods in the Phase II were added to get a better idea about the reproducibility (Fig.4.). The calculation of the throughput was added to the supplementary material.

- Analyses were cross checked in MetaboAnalyst. So all of the QC, OPLS-DA analyses, cross validation, etc were re-done in two different software packages?
- Yes, the data set for Phase III was evaluated with MetaboAnalyst software as well (data not shown). This was another verification step that lipidomic profiling is applicable for the differentiation of case and control samples and not biased by the operators.

REVIEWER COMMENTS

Reviewer #1 (Remarks to the Author):

The authors have improved the clarity of the text and the conclusion that lipidomics can differentiate PDAC vs. controls across measurements from multiple laboratories.

Reviewer #2 (Remarks to the Author):

The authors partially addressed my concerns. There are still somethings which I couldn't follow and are summarised as below:

1. I assume that figure 4 is generated based on the data from supplementary table 6a-6e. In supplementary table 6e and 6d the molar concentration of SM41:1 is below 1, which is very different from what is shown in Figure 4a
 2. For the phase II and phase III study, normalisation with NIST plasma has been applied. The concentrations of some lipid species are very different before and after normalisation. I wonder which data set are used for OPLS-DA?
 3. In supplementary table 6a, 6b and 6c, there are quite some identical values for different samples for a given lipid species, e.g. 0.0123 for TG 38:0 in table 6a, 1.1062 for PC 34:0 in table 6b etc. Can the authors explain?
 4. I wonder if the authors can comment on some big discrepancy of the concentrations of some lipid species between table 5a and 5b, e.g. TG50:1 in QC and some samples.
- One criterion to assess omic-based biomarker discovery study is whether the results can be reproduced by other labs. It is very good that the authors have provided very detailed description of the methods and including all the measurement in the supplementary tables. I recommend the authors carefully check the data in the supplementary tables.

Minor point:

There are no figure legend for Figure 3 c and d.

Reviewer #3 (Remarks to the Author):

To the authors:

- * The manuscript is much clearer and improved.
- * The authors have performed a good analysis with the software tools they have. The software apparently does not allow incorporation of additional covariates like sex that the authors show affect the results, forcing the authors to rely on subgroup analyses. Follow up work should include statistical modeling where everything can be put into one model.
- * line 234+: They state that no clustering was visible by stage. Is the ROC AUC the same by stage then too? That would be a more reliable assessment.
- * Figure 6C. Are the sens/spec/auc really all equal in the training set? same for the validation?
- * line 326+: I find it puzzling that the lipid markers useful for discrimination do not show any change after treatment or after surgery. That makes one really wonder the mechanism going on here, and what the markers are detecting. Are they really detecting disease? A marker of disease should be absent when no disease is present, so after surgery for sure the patients should look like controls. Though, it sounds like they only looked at a few lipids here. It would be better to apply the full score, output the score per person, and compare that pre/post surgery or treatment.
- * line 350+: The number of lipids tested for association with survival should be stated. I believe all lipids were tested so multiple comparison correction such as FDR is needed for the survival models. Better than that though, rather than testing association with survival for each lipid, it would be more informative and relevant to point of the paper to use the MDS score output in a Cox model (or similarly cut that score at the median or at tertiles and use KM). Also, this means only a couple of

scores are being tested (1 male, 1 female) reducing multiple comparison problems. Not sure why both KM and Cox models were used since the data was dichotomized at the median and therefore they would give identical results with the logrank test.

* line 422+ in Discussion: if the altered lipid metabolism originates from organs affected by pdac metastatic spread, this would diminish the ability for early detection (before metastasis).

* The authors should add to the discussion the implications on the conclusions that can be drawn since 67% of the lab 1 phase III samples were also included in phase II.

Response to reviews - NCOMMS-21-00608A, 2. revision

First of all, the authors would like to thank the reviewers once more for their careful feedback and comments, which helped to improve the quality of our manuscript. All comments and concerns were carefully studied, appropriate changes implemented in the revision or explained in detailed point-by-point responses here.

Reviewer #1:

The authors have improved the clarity of the text and the conclusion that lipidomics can differentiate PDAC vs. controls across measurements from multiple laboratories.

We thank Reviewer #1 for the constructive feedback, which strengthened the manuscript and the positive evaluation.

Reviewer #2:

The authors partially addressed my concerns. There are still somethings which I couldn't follow and are summarised as below:

1. I assume that figure 4 is generated based on the data from supplementary table 6a-6e. In supplementary table 6e and 6d the molar concentration of SM41:1 is below 1, which is very different from what is shown in Figure 4a

The reviewer is right that the generated boxplots are based on the data presented in Supplementary Tables 6a-6e. However, the normalized data to the NIST plasma were used for boxplots. The data together with the calculation can be found below the summary table of absolute concentrations and below the summary table of statistical parameters. Therefore, it is needed to scroll down in this large Excel file to find the normalized data. The supplementary table guide was modified for better orientation in Excel sheets. Here we show boxplots for absolute concentrations (a) and also normalized concentrations to the NIST plasma (b) for SM 41:1. The normalization improves the comparability of concentrations among various methods, instruments, and laboratories, which is important for clinical applications.

2. For the phase II and phase III study, normalisation with NIST plasma has been applied. The concentrations of some lipid species are very different before and after normalisation. I wonder which data set are used for OPLS-DA?

The normalization with the NIST plasma was applied for the comparison and illustration of lipid concentrations obtained for different methods and laboratories. However, the absolute concentrations without normalization were used for OPLS-DA models in Phase II and III (Figure 5 and 6). In principle, both data with or without normalization can be used for construction of OPLS-DA models with similar conclusions. However, some lipids have to be excluded in case of data normalization to NIST plasma, because the literature values of some lipid species are not available, which subsequently reduced the range of lipid concentrations from our data sets. For this reason, we prefer to use the data set without normalization for constructing OPLS-DA plots.

3. In supplementary table 6a, 6b and 6c, there are quite some identical values for different samples for a given lipid species, e.g. 0.0123 for TG 38:0 in table 6a, 1.1062 for PC 34:0 in table 6b etc. Can the authors explain?

The described situation is explained by zero filling. It is common for omics techniques that some values for certain percentage of samples is not obtained due to various reasons, which should result in some zero values in the data sets. However, zero values are strongly discouraged for MDA models, because this could strongly affect them, and the use of some suitable zero filling approach is recommended. Based on our previous optimization, we replace zero values by 80% of the minimum detected concentration considering all samples. As a consequence, some lipid species show repeated concentration values, as noticed by the reviewer.

4. I wonder if the authors can comment on some big discrepancy of the concentrations of some lipid species between table 5a and 5b, e.g. TG 50:1 in QC and some samples.

The difference between Table 5a (UHPSFC/MS; TG 50:1 (mean): 209 nmol/mL) and 5b (Shotgun-MS TG 50:1 (mean): 25 nmol/mL) is the applied method for lipid quantitation in Phase I. Individual MS

based approaches differ in the sensitivity for some lipid classes, for example UHPSFC/MS is highly sensitive for TG, while shotgun has the worse sensitivity for TG due to the absence of specific transition for TG. This could result in big discrepancies between values for TG and some other lipid classes between these data sets. Furthermore, Phase I was the starting stage of this research, and improvements were made Phase II and Phase III due to the method optimization and validation, but the principal differences between UHPSFC/MS and shotgun MS methods cannot be completely excluded, but only diminished. As discussed above, the normalization could significantly reduce such discrepancies for inter-method and inter-laboratory comparison, as shown below on the example of molar concentrations (a) without normalization and (b) with normalization.

One criterion to assess omic-based biomarker discovery study is whether the results can be reproduced by other labs. It is very good that the authors have provided very detailed description of the methods and including all the measurement in the supplementary tables. I recommend the authors carefully check the data in the supplementary tables.

We fully agree with the reviewer that reproducibility is the unmet need for the acceptance of clinical lipidomics. For this reasons, we follow FDA and EMA guidelines in our quantitative workflows, such as the use of internal standards, validated MS methods, and the measurement of QC samples as well as a reference sample (like the NIST 1950 plasma, allowing normalization), as reported in our previous works. The supplementary tables were checked at least 3 times, so we believe that they are free of errors. Based on these tables, statistical models were created, so this is an additional verification to exclude hidden typing errors.

Minor point:

There are no figure legend for Figure 3 c and d.

The authors are grateful for the careful checking. The figure legend was corrected.

Reviewer #3:

* The manuscript is much clearer and improved.

1. The authors have performed a good analysis with the software tools they have. The software apparently does not allow incorporation of additional covariates like sex that the authors show affect the results, forcing the authors to rely on subgroup analyses. Follow up work should include statistical modeling where everything can be put into one model.

The reviewer is right that we have done our best within the limits of the software that we are using for the statistical analysis. We observe small improvement of statistical parameters for all data sets, if the gender separated models are used, therefore we use them throughout the whole work. The change of statistical modeling into one model approach would probably require a different software tool, which may result in the comprehensive revision of the whole concept. Anyway, we agree with the reviewer suggestion to focus on this aspects in the follow up work.

2. line 234+: They state that no clustering was visible by stage. Is the ROC AUC the same by stage then too? That would be a more reliable assessment.

The influence of stage on the lipid concentrations were further investigated for Phase III by box plots for selected lipid species (Figure 7b and c, see 276+). The ROC curves and AUC values considering only stages T1 + T2 and T3 + T4 are illustrated here (all samples - training and validation set and both genders where the T stage information was available were included - healthy controls (N) = 262, case samples: T1/T2 = 54; T3/T4 = 350). The stage does not show any impact on the prediction, even though the sample number for T1/T2 was relatively low due to limited availability of early stage PDAC patients. The ROC graph supports our claim that the lipidomic profiling is capable to differentiate early stage PDAC from control samples.

3. Figure 6C. Are the sens/spec/auc really all equal in the training set? same for the validation?

The authors are grateful for the careful checking of the data from the reviewer. The authors reassessed the data again and got the same result, as illustrated below:

Phase III - females		Training set			Validation set		
Prediction	Sample type	Samples	Error	[%]	Samples	Error	[%]
Sensitivity	Case (T)	211	12	94.31	10	2	80.00
Specificity	Control (N)	124	7	94.35	60	12	80.00
Accuracy	Sum	335	19	94.33	70	14	80.00

4. line 326+: I find it puzzling that the lipid markers useful for discrimination do not show any change after treatment or after surgery. That makes one really wonder the mechanism going on here, and what the markers are detecting. Are they really detecting disease? A marker of disease should be absent when no disease is present, so after surgery for sure the patients should look like controls. Though, it sounds like they only looked at a few lipids here. It would be better to apply the full score, output the score per person, and compare that pre/post surgery or treatment.

Of course the knowledge of disease mechanism is extremely important task, but this is not trivial and will require well planned biological models, e.g., PDX or humanized mice. Such experiments are planned within our upcoming project, but it will require additional few years of work. It is true that lipid profiles do not return quickly after surgery or treatment, which is independent whether the lipid profile (Figure 8a – OPLS-DA) or individual markers (Figure 8b and c – boxplots) are investigated. The basic issue is that the surgical removal of pancreatic tumor cannot be simply considered as the healthy state, because the return of disease is very common for PDAC, which also fits with very low survival of PDAC patients (<10%), regardless of surgery or treatment. The changed lipidomic profile of blood may be caused by the systematic response of organism, not related only to tumor itself, because a small tumor in stage T1 could hardly affect the whole blood in the organism so significantly that it could be analytically detectable. Our results do not provide the mechanism so far, but what seems likely is that that the surgical removal of the tumor does not mean the subject is cured, which is in agreement with the real outcomes of PDAC patients. In any case, this problem is complex, and we would rather not claim some unconfirmed hypothesis in the manuscript without the strong support of experimental data.

5. line 350+: The number of lipids tested for association with survival should be stated. I believe all lipids were tested so multiple comparison correction such as FDR is needed for the survival models. Better than that though, rather than testing association with survival for each lipid, it would be more informative and relevant to point of the paper to use the MDS score output in a Cox model (or similarly cut that score at the median or at tertiles and use KM). Also, this means only a couple of scores are being tested (1male, 1female) reducing multiple comparison problems. Not sure why both KM and Cox models were used since the data was dichotomized at the median and therefore they would give identical results with the logrank test.

For the Kaplan-Meier survival analysis and the Cox Hazard proportional analysis, lipid concentrations were converted into binary code. Therefore, the median concentration of the lipid species for all samples were calculated, and individual lipid species concentrations were classified to 0, when the concentration was smaller than the median concentration of all samples or 1, when the concentration was bigger than the median concentration of all samples. The concentration tables normalized to the NIST plasma were used. All lipid species, the gender and CA19-9 were converted to binary code for the Kaplan-Meier survival analysis for all methods applied in Phase II (Number of lipids tested separately- UHPSFC/MS (Method 1): 175; shotgun MS (LR) (Method 2): 165; shotgun MS (HR) (Method 3): 151; and RP-UHPLC/MS (Method 4): 232). The Kaplan-Meier plots and the logrank test were then generated in R, and results are summarized in Supplementary Table 13. The Kaplan-Meier analysis represents a univariate analysis method, which does not work well for quantitative measures, so the conversion was necessary. The Cox Hazard analysis works well for both univariate and quantitative approaches. The Cox proportional regression analysis was used to simultaneously assess the effect of several lipid species and CA19-9 on the survival time probability. The investigated lipid species for the Cox-Hazard analysis were designated by determining the statistically significant lipid species according to the KM p-value for at least 3 out of the 4 methods applied in Phase II.

6. line 422+ in Discussion: if the altered lipid metabolism originates from organs affected by pdac metastatic spread, this would diminish the ability for early detection (before metastasis).

The sentence was reformulated.

7. The authors should add to the discussion the implications on the conclusions that can be drawn since 67% of the lab 1 phase III samples were also included in phase II.

Phase II aimed the independent evaluation that lipidomic profiling is capable for the differentiation of case and control samples with various methods and laboratories. The reviewer is right that there is the sample overlay for Phase II and Phase III. However, the purpose of both phases differ and is not in contrast to each other. A clarification was added to the discussion to make this fact clear to readers.

REVIEWER COMMENTS

Reviewer #2 (Remarks to the Author):

I read the revised manuscript with enthusiasm, unfortunately I didn't find the revised version to be an improvement. The supplementary Table numbering no longer match the citation in the result section. I just list a couple of examples:

1. Line 102-103, the text indicates that "Lipid species were quantified by using exogenous lipid class internal 103 standards (IS) added to the serum before the sample preparation (Supplementary Table 3)". In the new version of supplementary Tables, there is no Supplementary 3. Instead, the IS information is now in Supplementary Table 1.

2. Line 125-135, the text mentioned supplementary Table 5a, 5b and 5c. Again, there are no Supplementary Table 5a, 5b and 5c.

In the rebuttal letter, the authors said that the supplementary tables were checked at least 3 times. However, I recommend the author to check once more the text and also the text in "Overview of Supplementary tables".

Response to reviews

The authors are grateful for the careful reading and evaluation of the manuscript by the reviewer and the editor.

Reviewer #2 (Remarks to the Author):

I read the revised manuscript with enthusiasm, unfortunately I didn't find the revised version to be an improvement. The supplementary Table numbering no longer match the citation in the result section. I just list a couple of examples:

- The authors are very sorry for the mismatch! During the second revision, several changes of the Supplementary Table file according to the advices of reviewers were made. Unfortunately, the wrong (old) Supplementary Table file was uploaded for the revision. We are very grateful for discovering the mismatch by the reviewer. It is corrected now.

1. Line 102-103, the text indicates that "Lipid species were quantified by using exogenous lipid class internal 103 standards (IS) added to the serum before the sample preparation (Supplementary Table 3)". In the new version of supplementary Tables, there is no Supplementary 3. Instead, the IS information is now in Supplementary Table 1.

2. Line 125-135, the text mentioned supplementary Table 5a, 5b and 5c. Again, there are no Supplementary Table 5a, 5b and 5c.

In the rebuttal letter, the authors said that the supplementary tables were checked at least 3 times. However, I recommend the author to check once more the text and also the text in "Overview of Supplementary tables".

- The Supplementary Table file was significantly edited according to the instructions of the editor. Supplementary Tables fitting to an A4 page were moved to the Supplementary Information file, and Supplementary Tables bigger than A4 were transformed to Supplementary Data Tables. As a consequence, the whole numbering of the Supplementary tables are changed.